# Bimodality and alternative equilibria do not help explain long-term patterns in shallow lake chlorophyll-a

Thomas A. Davidson [1,2] ✉, Carl D. Sayer[3], Erik Jeppesen[1,2,4,5,6], Martin Søndergaard[1,4], Torben L. Lauridsen[1,2,4], Liselotte S. Johansson[1], Ambroise Baker [7] & Daniel Graeber [8] ✉

Since its inception, the theory of alternative equilibria in shallow lakes has evolved and been applied to an ever wider range of ecological and socio-ecological systems. The theory posits the existence of two alternative stable states or equilibria, which in shallow lakes are characterised by either clear water with abundant plants or turbid water where phytoplankton dominate. Here, we used data simulations and real-world data sets from Denmark and north-eastern USA (902 lakes in total) to examine the relationship between shallow lake phytoplankton biomass (chlorophyll-a) and nutrient concentrations across a range of timescales. The data simulations demonstrated that three diagnostic tests could reliably identify the presence or absence of alternative equilibria. The real-world data accorded with data simulations where alternative equilibria were absent. Crucially, it was only as the temporal scale of observation increased (>3 years) that a predictable linear relationship between nutrient concentration and chlorophyll-a was evident. Thus, when a longer term perspective is taken, the notion of alternative equilibria is not required to explain the response of chlorophyll-a to nutrient enrichment which questions the utility of the theory for explaining shallow lake response to, and recovery from, eutrophication.

The idea that ecosystems can have markedly different alternative equilibria under similar environmental conditions has its historical roots in theoretical ecology[1] and the concept of the catastrophe fold[2]. Evidence for this theory gained credence from empirical data reporting catastrophic regime shifts across a range of ecosystems, including deserts[3], oceans[4], and coral reefs[5,6]. The terminology describing these nonlinear changes evolved to regime shift[7] and later critical

transitions[8], but the kernel of the idea remains the same: An ecosystem can occupy one or more alternative states or equilibria over a relatively broad range of environmental conditions, with each respective state characterised by stability in the face of environmental change due to resistance engendered by a number of feedback mechanisms[9].

The existence of alternative equilibria, both in theoretical models and real-world examples, has been the subject of debate[7,10–15]. In fresh

[1]Lake Ecology, Department of Ecoscience and Arctic Research Centre, Aarhus University, Aarhus, Denmark. [2]WATEC Aarhus University Centre for Water Technology, Aarhus University, Aarhus, Denmark. [3]Environmental Change Research Centre, Department of Geography, University College London, Gower Street, London, WC1E 6BT, UK. [4]Sino-Danish Centre for Education and Research (SDC), University of Chinese Academy of Sciences, Beijing, China. [5]Limnology Laboratory, Department of Biological Sciences and Centre for Ecosystem Research and implementation, Middle East Technical University, Ankara, Turkey. [6]Institute of Marine Sciences, Middle East Technical University, Erdemli-Mersin, Turkey. [7]School of Health and Life Science, & National Horizons Centre, Teesside University, Middlesbrough TS1 3BX, UK. [8]Aquatic Ecosystem Analysis, Helmholtz-Centre for Environmental Research – UFZ, Brückstr. 3a, 39114 Magdeburg, Germany. ✉e-mail: thd@ecos.au.dk; daniel.graeber@ufz.de

waters, a review of the evidence of regime shifts found that of the 135 reported, very few met the criteria of Lees, Pitois[16] and Peterson[17] for a true regime shift, suggesting that the phenomenon is not common in nature[14]. Most recently, in a large meta-analysis, ref. [15] found that threshold transgressions (or catastrophic changes) were rarely detectable and that ecological response was most commonly characterised by progressive change.

Shallow lakes are generally accepted as the best example of an ecosystem occupying alternative equilibria[14,18–20]. Here, in place of a predictable, deterministic relationship between nutrients and phytoplankton biomass (chlorophyll-a), there exist two alternative equilibria, each state being reinforced by a suite of positive feedback mechanisms[9]. Specifically, over a given nutrient range, (for example, 50 and 200 $\mu$g l$^{-1}$ TP), shallow lakes should have either clear water and abundant submerged plants or turbid water with a dense phytoplankton crop where submerged plants are absent and in principal there should be no intermediate state[9]. There are, however, a growing number of observations showing marked inter-annual variability in the relative dominance of phytoplankton over submerged plant abundance[21–27]. As is often the case when theory confronts data[28] the theory has undergone some revisions, including multiple stable states[29] cyclical shifts between states[30] and the presence of the ghosts of long-term transients[24]. In addition, other studies have identified the regular coexistence of clear water and turbid conditions within a single growing season[31,32]. Thus, the stability of each respective state over both short and longer time periods has become increasingly questionable and thus worthy of examination.

Uncertainty over the identification of alternative equilibria and regime shifts may result from mismatches in the scale of investigation[10,14], where the mechanisms shaping ecological state operate on different temporal scales to the observations[33]. To understand ecosystem change over time in a dynamic and variable world, it is vital that the scale of observation matches that of underlying ecological processes[33]. Recent work using data from the Long-Term Ecological Research network demonstrated that an insufficiently long temporal perspective can result in spurious conclusions[34]. In particular, it may be the absence of a sufficiently long-term view of systems characterised by inter-annual variability that makes it difficult to define an ecosystem state as stable or transient. Long-term studies of shallow lakes have tended to show gradual rather than step-changes in ecological conditions in response to reductions in nutrient concentration[15,35,36]. Although it has been suggested that regime shifts may occur slowly for some ecosystems[37], regime shifts in shallow lakes have been characterised as catastrophic and occurring over short time scales[7].

Here, we examine how shallow lakes respond along a gradient of nutrient enrichment using data covering a range of temporal scales, from a single-year to 5-year means. For this range of temporal scales, we aim to robustly test the presence or absence of alternative equilibria, assessing the utility of the idea versus the previously held notion of gradual, linear ecosystem response in explaining observed patterns. In short, we pose the question: Does the theory of alternative stable states stand the test of time? The multi-year perspective taken here to investigate how nutrients shape chlorophyll-a reveals that as, the temporal scale of observation lengthens, a strong deterministic relationship between nutrients and chlorophyll-a is revealed, and ideas of bimodality and alternative equilibria are in no way useful in explaining the patterns in the data.

## Results

The data consisted of 902 lakes with 2986 summer mean (May to September) observations from Denmark and North America covering a range of sizes and levels of nutrient enrichment (Table 1). The concept of the analyses centred on the comparison of single-year observations with multiple-year means of total phosphorus (TP), total nitrogen (TN)

and chlorophyll-a, with increasing numbers of years included in the calculation of multiple-year means up to a five-year mean conducted in an identical manner for both simulated and real-world data.

To ensure a robust basis with which to compare the analysis based on a single year with those based on equivalent multi-year means, the real-world and simulated datasets were sampled identically using a hierarchical bootstrap procedure. Here the real-world and simulated datasets were sampled 1000 times with replacement for the lakes and then single or multi-year means of the key variables (TN, TP and chlorophyll-a)derived without replacement[38] (see methods and supplementary materials for details). For each of the 1000 random samples per bootstrap analysis, GLM models were constructed. However, the GLM models did not always converge and in the event of nonconvergence, this random sample was not used. This resulted in $n = 608$, $n = 696$ and $n = 633$ random samples for 1, 3 and 5-year single year data, respectively and $n = 584$ and $n = 795$ for the 3 and 5-year mean data, respectively. For the simulations, $n = 1000$ for most scenarios (models converged for all iterations), with the exception of the 1-year simulations with ASS where $n = 945$, and without ASS where $n = 864$. The distributions of the data are shown as density plots, with the error bars being the highest density interval with a 95% credible interval.

Details of the results of the data simulations and the four different scenarios of alternative equilibria are given in the methods and supplementary materials. The four simulation scenarios covering the presence, absence and different constellations of alternative equilibria demonstrated that, as the temporal scale of the dataset lengthens from a single year to the five-year mean, the diagnostic indicators (Fig. 1) have divergent responses that reflect the presence or absence of alternative equilibria in a given dataset (Supplementary Figs. 3, 4). The presence of alternative equilibria was reflected by characteristic changes in the three indicators as the number of years used to calculate the mean increased. These changes reflecting the presence of alternative equilibria were, no increase in the $R^2$ of the linear model between nutrients and chlorophyll-a and increasing bimodality in both the residual patterns and the kernel density plot (Fig. 1 and Supplementary Fig. 3). The data simulations reveal that the diagnostic indicators can not only identify datasets with a total absence or total dominance of alternative equilibria, but also that they are sensitive to different scenarios where some of the lakes contained both states within a times series (See Supplementary Notes 2). In this latter scenario, there was no increase in $R^2$ and an increase in the bimodality of the residuals of the model, albeit less pronounced than in scenario 4 (Supplementary Fig. 4). The diagnostic tests were also robust to datasets with occasional unstable alternative states, in this scenario with occasional instability the pattern in the diagnostic tests was most similar to the scenario where alternative equilibria were absent (Supplementary Fig. 3). Thus, the combination of the four scenarios and the three diagnostic indicators provide a well-defined set of characteristics which can be used to identify either the presence or the absence of alternative equilibria in a given dataset. In this way, we avoid the dangers of a conjunction fallacy[39] where the assertion that the real-world dataset does not contain alternative equilibria is based on the fact that there is no strong evidence of its presence, rather than a pattern that confirms its absence.

The relationship between nutrient concentration and chlorophyll-a for single-year observations and for the means of multiple years' data (Table 2 and Fig. 2) show that TP generally explained more variance than TN, with the variance explained by TP actually similar to that explained by TN and TP combined. The models based on the means from multiple years' data always explained more variance in the chlorophyll-a compared with those based on the comparable single year's data (Table 2). Furthermore, lengthening the period of time on which the multi-year means were based increased the explanatory

**Table 1 | Number of lakes, means (1 SD; minimum-maximum) of the contemporary data from the USA, Denmark and the full dataset**

| Variable | USA | Denmark | All data |
|---|---|---|---|
| Number of lakes | 122 | 780 | 902 |
| Mean depth (m) | 2.0 (0.7; 0.4–3.0) | 1.1 (0.7; 0.0–3.0) | 1.2 (0.8; 0.0–3.0) |
| Area (ha) | 150.4 (389.0; 1.7–3179.2) | 34.2 (114.0; 0.0–1713.0) | 49.9 (182.1; 0.0–3179.2) |
| Chlorophyll-a ($\mu$g L$^{-1}$) | 37.0 (44.2; 0.3–362.5) | 63.8 (65.5; 0.0–520.1) | 57.3 (62.1; 0.0–520.1) |
| Total nitrogen (mg L$^{-1}$) | 1.4 (1.1; 0.1–5.9) | 1.7 (1.0; 0.3–6.0) | 1.6 (1.0; 0.1–6.0) |
| Total phosphorus ($\mu$g L$^{-1}$) | 87.8 (84.9; 4.6–516.9) | 179.2 (194.0; 4.4–1457.9) | 157.2 (178.3; 4.4–1457.9) |

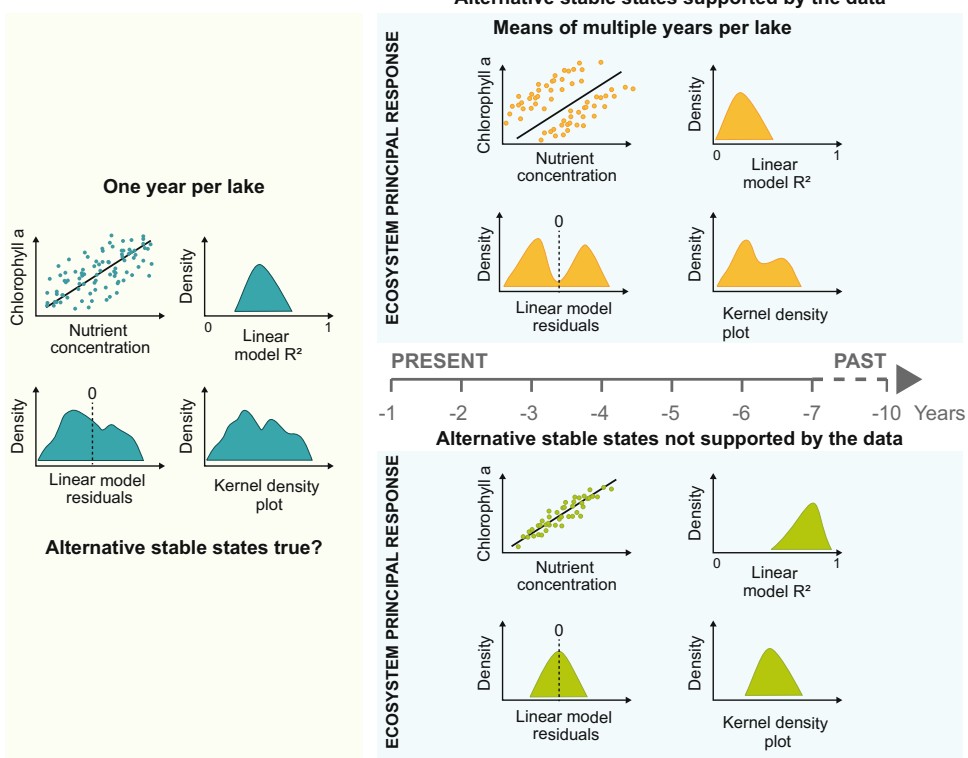

**Fig. 1 | Detecting alternative equilibria in spatial data at different temporal scales.** The temporal scale increases from a single year on the left to the decadal scale on the right. The data shows typical patterns of nutrient and chlorophyll-a found in shallow lakes for a single year, with results of the three diagnostic tests (see methods) applied to detect the presence or absence of alternative stable states (ASS). Single-year data does not provide strong evidence for or against the existence of ASS. In addition to examining the patterns in the scatter plots of nutrients vs chlorophyll-a, the tests are (i) the $R^2$ of the model, (ii) the residuals of a linear model of nutrients vs chlorophyll-a, (iii) Kernel density plots of chlorophyll-a data. We present expected patterns (derived from simulations) that would suggest the presence (above the line) or absence (below the line) of ASS in shallow lakes. As the temporal scale of the observations increases from single-year data to multiple-year means the expectation is that inter-annual variability should even out and the presence or absence of ASS should become apparent in the proxies (above the horizontal line). The scatter plot of chlorophyll-a versus the nutrients will show two clouds of data (turbid or clear), hence, (i) the $R^2$ of the model will decrease, as a single linear model cannot predict two alternatives, (ii) the residuals of a linear model correlating nutrients to chlorophyll-a will show multimodality and (iii) the kernel density plot of chlorophyll-a will deviate from unimodality. Conversely, in the absence of ASS (below the horizontal line), the link between nutrient concentration and chlorophyll-a becomes increasingly well predicted by a linear model, resulting in a larger $R^2$ with an increasing number of averaged year, unimodality of the model residuals and of the kernel density function.

power of the models, resulting in over 80% of the variance in chlorophyll-a being explainable when a 5-year mean of the data was used (Table 2). Despite this general high degree predictability, there were some 'outlier' type observations even at the five-year mean temporal scale where there were high or low chlorophyll-a values for a given nutrient value (Fig. 2). These can be explained when the N:P ratio is considered as this can affect the relationship between the individual nutrient (TP or TN) and chlorophyll-a. For example, a lower chlorophyll-a value for a given TP may be due to nitrogen limitation rather than reflecting alternative equilibria (See Supplementary Figs. 1, 2).

For the real-world data, the patterns in the diagnostic tests for the presence and absence of alternative equilibria covering the range of time scales from a single year to the 5-year mean, show very close accord with the data simulation where alternative stable states are absent (Fig. 3). This is characterised by increasing $R^2$ as the number of years in the mean increases, unimodal residual patterns and a unimodal kernel density plot that does not change as more years are added to the mean. Furthermore, as was the case in the simulations, the multi-year mean models always had a higher $R^2$ when compared with their single-year equivalent (Fig. 3). The data were reanalysed for both a limited range of nutrients and also for each region (Denmark and the

**Table 2 | Summary statistics of the GLM modelling of nutrient and chlorophyll-a relationships based on the data sampling by the hierarchical bootstrap for TP, TN and TN + TP models (mode (in bold) and 95% highest density interval)**

| Averaged years | Number of lakes | TP | | | TN | | | TP & TN | | |
|---|---|---|---|---|---|---|---|---|---|---|
| | | Single year $R^2$ | Mean years $R^2$ | % change | Single year $R^2$ | Mean years $R^2$ | % change | Single year $R^2$ | Mean years $R^2$ | % change |
| 1 | 902 | **0.44** (0.38–0.5) | | | **0.35** (0.3–0.4) | | | **0.46** (0.4 –0.51) | | |
| 2 | 203 | **0.64** (0.53–0.72) | **0.72** (0.61–0.77) | 12.5 | **0.41** (0.31–0.51) | **0.49** (0.39–0.58) | 19.51 | **0.63** (0.54–0.72) | **0.72** (0.62–0.77) | 14.29 |
| 3 | 142 | **0.68** (0.55–0.76) | **0.78** (0.66–0.84) | 14.71 | **0.47** (0.32–0.55) | **0.56** (0.46–0.67) | 19.15 | **0.70** (0.55–0.77) | **0.79** (0.66–0.85) | 12.86 |
| 4 | 124 | **0.63** (0.52–0.75) | **0.75** (0.65–0.85) | 19.05 | **0.45** (0.32–0.56) | **0.61** (0.51–0.71) | 35.56 | **0.63** (0.51–0.76) | **0.78** (0.63–0.85) | 23.81 |
| 5 | 99 | **0.7** (0.58–0.78) | **0.83** (0.77–0.87) | 18.57 | **0.51** (0.38–0.63) | **0.69** (0.60–0.77) | 35.29 | **0.71** (0.60–0.79) | **0.84** (0.79–0.88) | 18.31 |

Results of single-year data analysis are compared with multiple mean year data for increasing numbers of averaged years for models based on TP, TN and TP+TN and the percent change of $R^2$ of single years versus averaged years is given.

USA) separately (see SI for details and results). There were some differences in the results of these smaller datasets but the overall patterns agreed with the larger dataset in identifying the absence of alternative equilibria in the real-world datasets.

## Discussion

The use of multiple-year means to interrogate the relationship between nutrients and chlorophyll-a in shallow lakes reveals that, as a longer-term perspective is taken, a predictable linear relationship between nutrient concentration and chlorophyll-a emerges, with over 80% of the variance explained for the 5-year mean data (Figs. 2, 3 and Table 2). In common with this study, most previous investigations from shallow lakes using a single-year time scale show a weak albeit significant relationship[27,40] between nutrients and chlorophyll-a (Fig. 2). This cloud of chlorophyll-a data in response to TP or TN, typically found for single-year data, does not represent strong evidence for either the presence or absence of alternative equilibria in nature. In contrast, the analysis of simulated and real-world data covering a range of temporal scales, from single to multiple-year means, strongly indicates the absence of alternative equilibria in the large real-world dataset and that chlorophyll-a is, to a very large extent, determined in a highly predictable way by nutrient concentrations. TP was the best predictor of chlorophyll-a in all the permutations of the dataset and this was more marked in the reanalysis of the data with a restricted range of nutrient concentration (see Supplementary materials S3). In addition, there were also cases where the N:P ratio was a factor in determining the relationship between an individual nutrient concentration (i.e. TP or TN) and chlorophyll-a. However, these occasional outlier sites do not reflect alternative equilibria but instead the importance of both TN and TP in shaping chlorophyll-a concentrations (Supplementary Fig. 2)[41].

The absence of a strong relationship between nutrients and chlorophyll-a over a single year contrasts with that found for the multi-year mean data. This is, in part, the result of year-to-year variation in nutrient-chlorophyll-a relationships, which this analysis shows even out over time. The year-to-year variation is the result of a range of internal and external processes, which either maintain some disequilibrium or cause the inter-annual variation in nutrient-chlorophyll relationships. External factors can maintain persistent or transient disequilibrium conditions, for example, a persistent drying out which resets the system through changing sediment characteristics and by eliminating or reducing populations of fish favours clearer water[17,24]. Other processes are a mix of external drivers which vary between years and alter internal processes. An example of this is when periods of ice cover are longer, altering trophic interactions by favouring larger-bodied *Daphnia*, which increases grazing pressure on algae in spring resulting in clearer water the following year[42]. In agreement with recent longer-term studies, we find that single-year observations can give misleading results[34], perhaps as the scale of observations do not match the multiple-year scale at which the effects of eutrophication manifest themselves. This may be particularly true for ecological processes linked to the top-down control of trophic structure, which vary on longer time scales. For example, multi-year internal population dynamics between fish and zooplankton affect inter-annual variation in fish predation on zooplankton and thus, via the trophic cascade, produce large year-to-year variation in chlorophyll-a at a given nutrient level[43]. Indeed, the short-term successes of biomanipulation (the removal of zooplanktivorous or addition of piscivorous fish) in restoring clear-water conditions and abundant macrophytes in lakes is a testament to the strength of trophic cascades involving fish and cladoceran zooplankon. Scheffer, Hosper[9] used the dramatic results of biomanipulation to support the ideas of alternative equilibria. The authors did, however, sound a note of caution, stating that it would only be possible to test the veracity of the theory after a number of years when the persistence of clear-water conditions could be

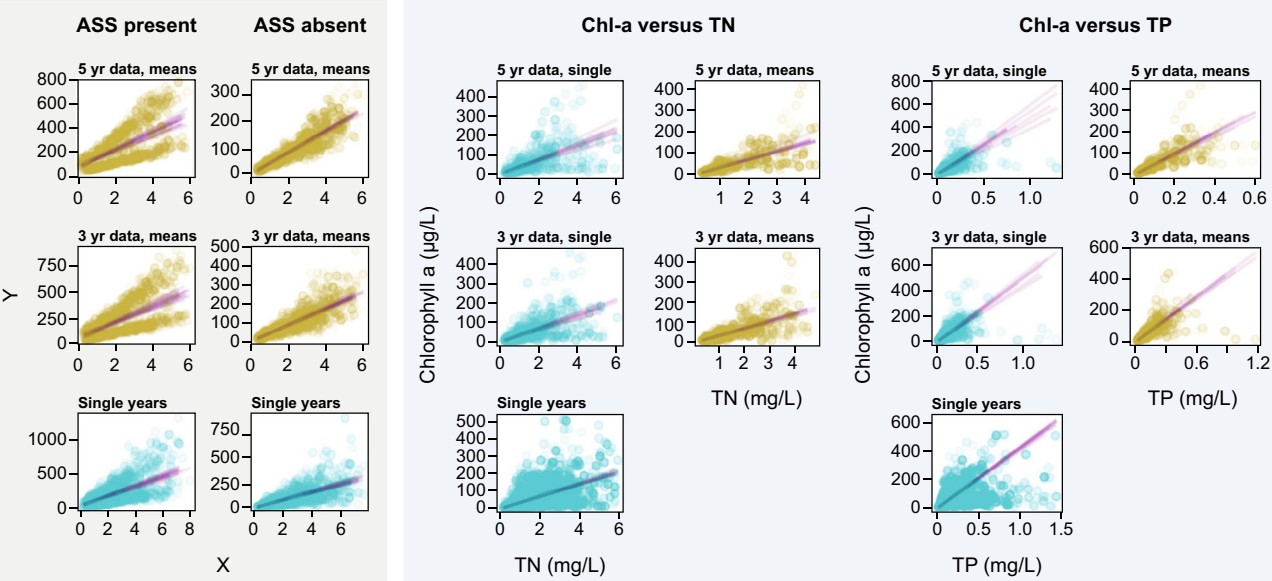

**Fig. 2 | Scatter plots of the results of simulated data with and without alternative equilibria.** There is an increasing temporal scale: from single-year observations to 3-year means and then 5-year means for the simulated data in the two panels on the left. Scatter plots of the TN and TP relationship with chlorophyll-a from real-world data with the different temporal perspectives of single-year data, 3-year means and 5-year means. The data were sampled 1000 times iteratively (see methods) for each given time period and a model was derived from each of the selections where the GLM converged (see results for numbers), providing a distribution of models and residuals for each temporal perspective. The shading of each point in the plots reflects the number of times it was selected in the random sampling procedure. The results show that as the temporal perspective lengthens the relationship between nutrients and chlorophyll-a becomes more predictable and that the models based on the multiple-year mean data always explain more variation than their single-year counterpart.

assessed. Evidence now available from multiple long-term monitoring datasets[19] and paleolimnological studies[44] shows that, where nutrient levels have not also been reduced, the positive effects of biomanipulation do not persist and there is typically a gradual return to turbid conditions over a few (5–10) years. Also, the response to nutrient loading reduction suggests a general gradual change, rather than an abrupt shift when a certain nutrient level is reached[15].

Our study does not refute the decades of research showing the influence of the trophic interactions in shallow lakes, where, for example, abundant submerged plants alter the relationship between nutrients and chlorophyll-a[40,45]. The suppression of phytoplankton by macrophytes can be both direct (e.g. allelopathy and reduced resuspension), and indirect via biotic interactions, with plant-associated zooplankton and invertebrates lowering the biomass of phytoplankton and periphyton[46,47], which in-turn improve conditions for plant growth. The plausibility of the idea of alternative equilibria in shallow lakes stems, in part, from the well-founded fact that the presence of abundant plants or dense phytoplankton can alter the food web and effect processes so as to promote their own success i.e. positive feedback. However, it is the stability of these feedbacks in the face of changing nutrient concentrations that is crucial to the idea of persistent or stable alternative equilibria and examining this stability is only possible with a longer-term perspective. The combination of simulated and real-world data here shows that, with a 3–5-year perspective, the top-down forces causing within[31] and between-year variation in the relationship between nutrients of chlorophyll-a[23] even out and the deterministic linear relationship between chlorophyll-a and nutrients becomes apparent and indeed even appears to hold over the entire length of the nutrient-enrichment gradient.

The combination of temporal perspectives provided here indicates that the mechanisms shaping the relationship between nutrients and chlorophyll-a in shallow lakes take multiple years to manifest their effects. In time-series data, this would typically result in a time-lagged ecosystem response to a press change, such as

relatively small changes in nutrient concentrations. Such lagged responses have been identified in other long-term studies of shallow lake ecosystems, with 10–15 year delays in the response of both fish[35] and submerged plants[48] to changes in the underlying nutrient concentrations. In shallow lakes, several factors have the potential to be slow-acting transients in the ecosystem. These include long-lived fish, the seed and propagule bank, along with sediment structure and sediment chemistry, each of which can potentially contribute to a disequilibrium between nutrients and chlorophyll-a as they hold a memory of conditions past. For example, changes in sediment chemistry, in particular organic content, which has the potential to adversely impact seed germination success[49] occur slowly over a number of years as a result of increased nutrient concentration. Legacy phosphorus in the sediment may also contribute to a lag in response where external nutrient inputs are reduced[50]. In addition, changes in sediment structure that make successful rooting of plants more difficult[51], take decades to occur, while long-lived and persistent propagule banks of aquatic plants[52] may take time to become buried and inactive. Thus, fish, macrophytes, sediment structure and chemistry appear to change at the decadal scale in response to nutrient level change and can contribute to a lag or a disequilibrium in ecosystem response to nutrient enrichment.

Shallow lakes are the ecosystem cited most often and with the most certainty as displaying alternative equilibria. The theory has developed over time to less easily testable ideas such as catastrophic regime shifts and critical transitions applied to an array of other ecological and socioecological systems[53]. The notions of critical transitions and alternative equilibria are so widely accepted that an area of research seeking to identify their early warnings has developed[54] although recent studies have found little convincing evidence for their existence in time series data[15]. As highlighted by Levin[33], it is vital that the time scale of observations match the time scale of the processes driving change, if we are to understand the temporal response of

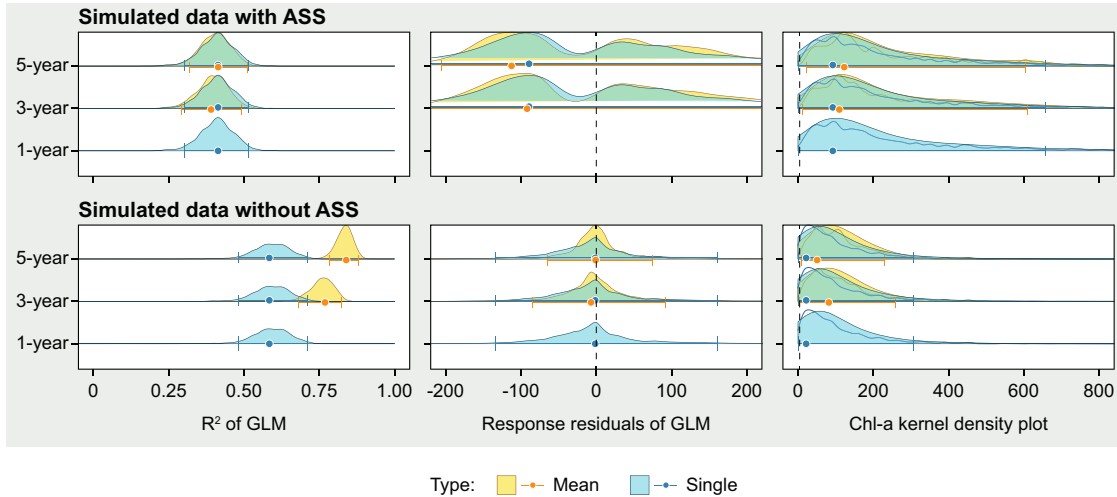

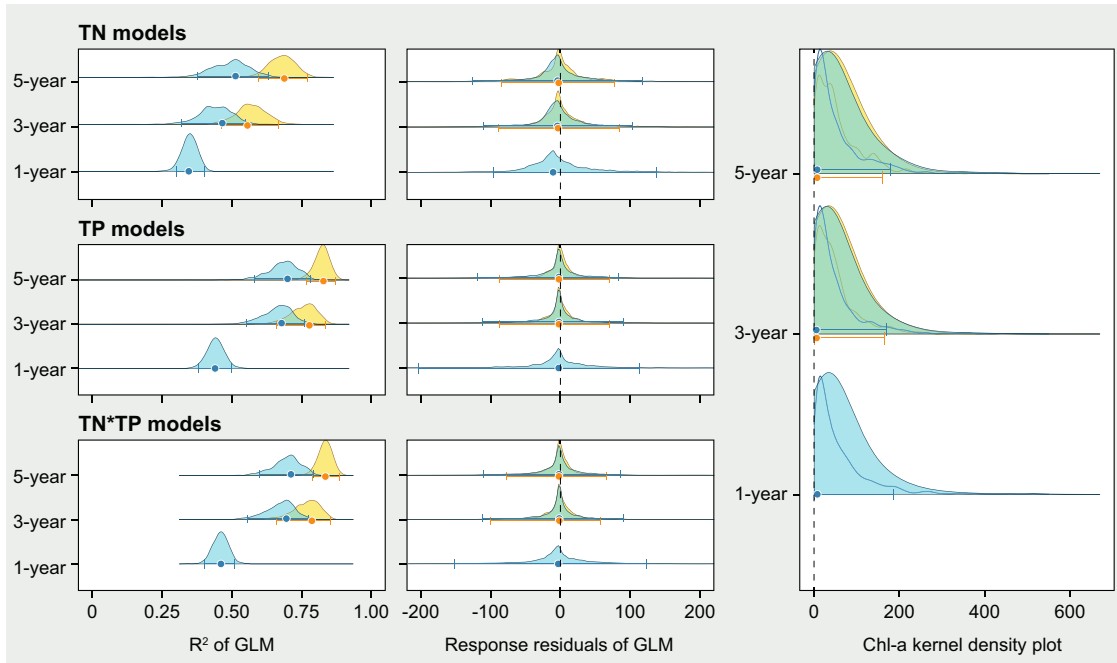

**Fig. 3 | Comparison of diagnostic tests for the presence or absence of alternative equilibria on simulated and real-world data at a range of time scales.** The figure shows the results of the three diagnostics tests (i) change in $R^2$, (ii) patterns in models residuals (single year, 3-year and 5-year means) and (iii) the kernel density plot (high (50 – filled) and low (10 – lines) bandwidth) for simulated datasets (**A**) with and without alternative equilibria and for the real-world data (**B**). Panel **A** demonstrates that there are characteristic patterns of the diagnostic tests for datasets but with and without ASS but that this requires an increase in temporal scale as the between-year variation obscures the patterns for single-year data. Panel **B** shows that real-world data follows the patterns of the simulated dataset with no alternative equilibria, with the $R^2$ of the models increasing, and the multi-year mean data always explain more data than their single-year equivalent, and no bimodality in the residual of the models or in the kernel density plot of chlorophyll-a data. About 1000 random samples were run per bootstrap analysis, for each of which we calculated GLM models. The final numbers vary slightly as the GLM models did not always converge and those which did not converge were not used (see results for numbers). The distributions of $R^2$, residuals and kernel density plots based on these multiple runs are shown as density plots, with the error bars being the highest density interval with a 95% credible interval.

ecosystems[34]. Here, we demonstrate that, as the time-window of observation lengthens, shallow lakes generally undergo a gradual change in response to eutrophication, which has been shown to involve a slow erosion of benthic habitats and pathways of production[55] and that nutrient levels overwhelmingly dictate chlorophyll-a concentrations. Whilst abundant macrophytes alter trophic interactions and nutrient to chlorophyll-a relationship in the short-term[27], the year-to-year variation evens out and bottom-up forces prevail in the longer term. Examining the patterns and variability of ecosystem change with the appropriate temporal perspective is vital to

better assess ecosystem change. The evidence presented here covers a range of time scales and shows that the ideas of bimodality and alternative equilibria do not help in explaining the patterns in the data and that, in short, the alternative stable states theory does not stand the test of time.

The implications for ecological theory and applied ecological management of this study are that there is a great benefit to be had from taking a longer-term perspective on the analysis of ecological change in the face of gradual changes in environmental conditions, such as climate and land use change. In terms of applied management,

our study highlights the pressing need to reduce nutrient inputs in shallow lakes to help maintain the resilience of their ecological processes and biodiversity in the face of rapid global change.

## Methods

### Real-world data

The dataset consisted of 2986 observations from 902 freshwater shallow lakes in Denmark and North America (data extracted from the LAGOSNE database on 22 February 2022 via R LAGOSNE package version 2.0.2)[56] (Supplementary Fig. 9). The Danish lakes were sampled for one or several years from 1984 to 2020 (data extracted in October 2021 from https://odaforalle.au.dk/main.aspx) (Supplementary Fig. 10). Prerequisites for inclusion in the analysis were that lakes had been sampled for physical and chemical variables at least four times or at least three times over the growing season (May to September) for the Danish or North American lakes, respectively, had a mean depth of less than 3 m and were freshwater. Water chemistry samples were analysed using standard methods and data for total phosphorus (TP), total nitrogen (TN) and chlorophyll-a are included here[57]. The mean and range of TP, TN and chlorophyll-a for the combined sites is given in Table 1, along with the values for each region separately.

To gain a longer-term perspective on the relationship between nutrients and chlorophyll-a, we calculated the across-year averages of the summer means of TP, TN and chlorophyll-a, sequentially increasing numbers of years included in the mean up to a total of a five-year mean, at which point there were only 99 lakes left in the dataset. In calculating the multi-year means we allowed a maximum gap of 2 years between observations (i.e. two observations could cover 3 years) to avoid including time series with too many missing years in between. Hence, only lakes with sufficient numbers of sequential data were included, resulting in a large drop in lake numbers as the length of the multi-year mean increased (Table 2).

### Numerical methods

**Diagnostic tests or proxies of alternative equilibria.** We modelled the response of chlorophyll-a to TP and TN using generalised-linear models[58] with Gamma distribution and an identity link on untransformed data for single-year and multiple-year means up to 5-year means. We used the Gamma distribution, as chlorophyll fit this significantly better than a normal or log-normal distribution. We used psuedo $R^2$ of the model along with the patterns of residuals, and finally, we plotted the kernel density of the chlorophyll-a values as diagnostics of the presence, absence or prevalence of alternative equilibria in the simulated and real work data.

To test how appropriate these diagnostics or proxies of alternative stable states in terms of how well they identify the existence of alternative stable states in randomly sampled multi-year data, we

1. Simulated two scenarios for the main manuscript, with and without alternative stable states in the data, which were as close to the real-life data as possible. The results of these scenarios appear in the main text (please see details below in the "Data Simulation" section).
2. We provide multiple scenarios with different degrees, or prevalence, of alternative stable states in the data, see simulations of alternative stable state scenarios. The results of these scenarios appear in Supplementary note 2.

### Hierarchical bootstrap approach

There are a large number of permutations of data, both real-world and simulated, that can provide a mean of the two to five sequential years from each lake in the time series data. It was vital to have a method that selects the data for analysis that provides a valid and comparable representation of both real work and simulated data and the models' errors. In order to provide this we used a non-parametric hierarchical bootstrap procedure[38]. The flowchart shows the data preparation and

data analysis steps of the hierarchical bootstrap procedure (Fig. 4). In the first step (step 1 in Fig. 4), all possible longer-term means are calculated for each lake. To keep as much data as possible, we decided to allow for up to 2 years of gap in the data between years. Taking the 5-year mean data as an example, if data from a lake existed for the years 1991 and 1994–1997, a 5-year mean would be calculated for the years 1991, 1994, 1995, 1996 and 1997. However, if the time series would contain a larger gap, e.g. data would only exist for the years 1991 and 1995–1998, no 5-year mean could be calculated. After the selection procedure, all the 2-year, 3-year and 5-year means are transferred into a new table (step 2 in Fig. 4).

The procedure is the same for each temporal scale from 2-year means to 5-year means. For the example of 5 mean years, lakes are randomly sampled from the full 5-year mean dataset in step 2 (Fig. 4) with replacement up to the number of lakes as in the original dataset, for the 5-year means 99 (step 3a). Here, the same lake can appear multiple times or not at all. This step is common for every bootstrap procedure[59]. However, since we have nested data (5-year means within lakes), we need a second step, in which for every resampled lake in step 3a, one 5-year mean is chosen (step 3b in Fig. 4). Then the three GLM models are produced from the randomly selected data in step 3c (Fig. 4). These steps are then repeated 1000 times to get a good representation of the uncertainties of the model. To ensure a fair comparison between single-year data and their equivalent multi-year mean data, we repeated the bootstrap procedure with single years only using only the lakes for which we also calculated multi-year means. To take the five-year mean as an example, there were 99 lakes where we could calculate at least one 5-year mean observation. First, we ran the bootstrap procedure to calculate 5-year mean values of TP, TN and chlorophyll-a (1000 times) and then took single years' values of TP, TN and chlorophyll-a (1000 times) from exactly the same 99 lakes. With this approach, exactly the same datasets with the same lakes and observations within lakes are used for the calculation of the multi-year means and their single-year counterparts, making for a robust analysis. The GLM models did not always converge. If either the TP, TN or TP*TN model with interaction did not converge, the iteration was not used in further analysis. The number of converging models equal for each iteration of random samples is given in the results.

The described hierarchical approach is the best way to reflect the structure of the original data. A simple, non-hierarchical bootstrap would favour lakes with more five-year means over lakes with fewer five-year means, simply because these make up a larger part of the data. Furthermore, sampling without replacement at the lake level would result in five-year means from lakes with few data dominating the produced random dataset, as every lake would be sampled every time, which then would result in high model leverage of 5-year means from lakes with less data. In contrast, the hierarchical procedure ensures that every lake has the same chance to end up in the randomly sampled bootstrap, in the second step, it ensures that of each sampled lake, every 5-year mean has the same chance to end up in the random dataset. These notions are in agreement with the findings of an assessment on how to properly resample hierarchical data by non-parametric bootstrap[38].

### Data simulation

**General approach of simulation assumptions and procedures.** We generated random scatter for the generalised-linear model based on Gamma distributions for two different "populations" of lakes with two different intercepts and slopes. At first, we calculated the linear equations for the two populations:

- Population i: $y_i = a_i + b_i * x$
- Population j: $y_j = a_j + b_j * x$

For each population $i$ and $j$, 99 samples (equalling the number of lakes in real-life data with 5-year means, $n_{lake}$) were generated with a

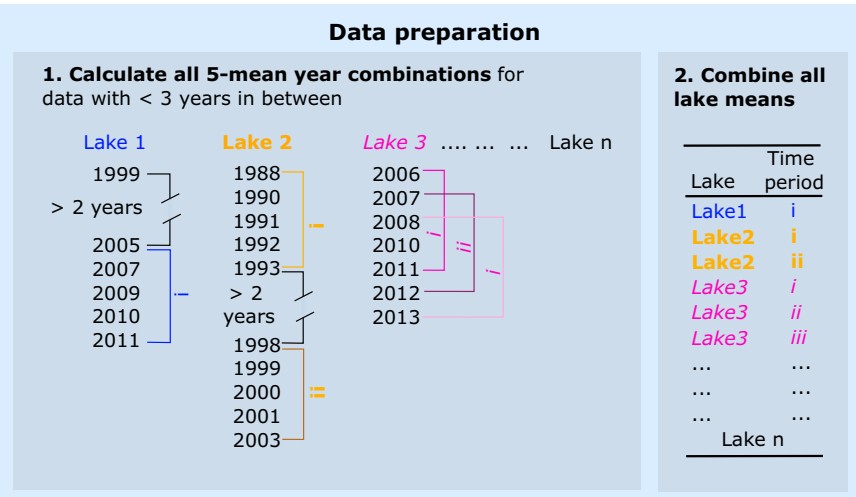

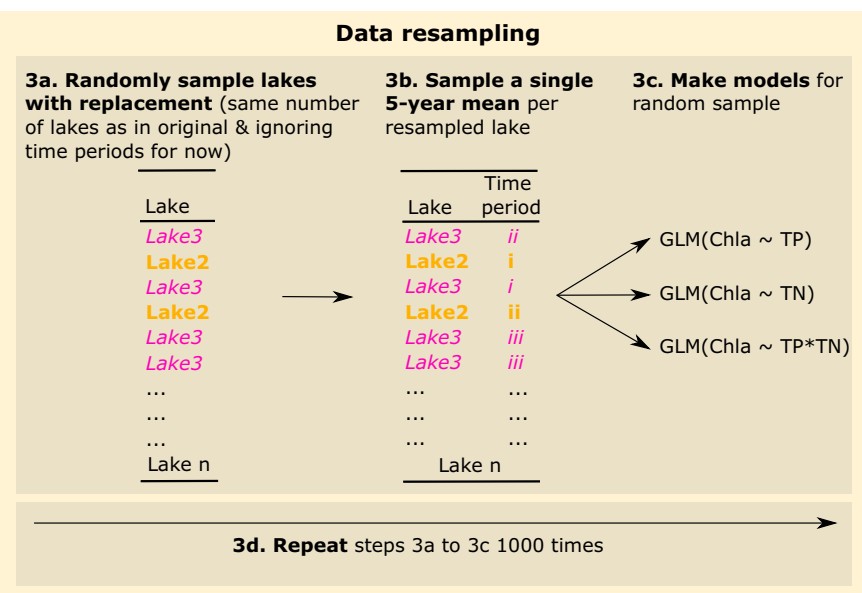

**Fig. 4 | Data preparation and analysis steps of the hierarchical bootstrap procedure.**

specific number of data points depending on the scenario ($n_{year}$) each, hence $n_{lake} = 1-99$ for each population of lakes, e.g. with 20 years ($n_{year} = 20$) each.

We found the real nutrient data to be normally distributed, with total nitrogen (TN) having a range between 0.33 and 4.93 mg/L and a constant coefficient of variation (CV, with a mean CV of 0.35) across this range (the same is true for total phosphorus (TP) at a shorter range). Hence, for each $n_{lake}$, the $x$ for the $n_{year} = 20$ were generated based on the mean range (mean per lake of the real-life data) and CV (0.35) from the real-life TN concentration data, hence with a range of 0.33 to 4.33 mg/L. Therefore the values and random variability of $x$ in the simulations are close to the true values of the TN concentrations. The $x$ is then fed into the linear equations above.

To the resulting $y_i$ and $y_j$ we added random noise based on the Gamma distribution (using the rgamma function in R). We used a Gamma distribution because the Chlorophyll-a concentration also follows a Gamma distribution. The variability of a Gamma distribution is expressed by the shape variable. The variability of chlorophyll-a, its shape value, equals 2.63. This shape value was used in the Gamma distribution of $y_i$ and $y_j$. The final calculated $y_i$ and $y_j$ had therefore a random rate calculated as shape/$y_i$ or shape/$y_j$. Hence, their variability in the y dimension was close to the true chlorophyll-a variability.

The data from both lake populations were then pooled and randomly sampled using the same hierarchical bootstrap procedure with 500 iterations for the scenarios in the supplementary materials and with 1000 iterations for main text simulation scenarios, which is identical to what was done for the real-world data.

**Simulation scenarios based on characteristics of real-world data.** The real-world 5-year mean data consisted of 99 lakes with 5–20 years of data for each lake. For the simulation scenario in the main text, we therefore randomly sampled between 5 and 20 data points for each of the 99 simulated lakes based on the $x$ distribution described above. Intercepts and slopes of the simulation, resembled the range of the true data (see scatter plots in Fig. 2 of the main manuscript).

In the alternative stable state scenario, we chose two slopes and intercepts for different populations of lakes:

Population i: $ai = 0$, $bi = 40$
Population j: $aj = 50$, $bj = 120$

We based the slopes and intercepts of the ASS scenarios on the diagnostic combination defined by Scheffer and Carpenter[7] which propose an abrupt shift in (a) the time series, (b) the multimodal distribution of states and (c) the dual relationship to a controlling factor. Here, the idea is that an ecosystem will jump from one state to the next

at the same (nutrient) conditions (different intercept and/or slope, condition a within ref. [7]), where any change in the nutrient will have different effects on algae or macrophytes (best represented by different slope, condition c), resulting in a multimodal distribution of the response (condition b). Hence, simulations are in line with what is predicted for ASS, but we took great lengths to also show other possibilities with the simulations in the Supplementary information to ensure we did not overlook any occasional occurrence of alternative equilibria.

Here, the appearance of alternative stable states in the data could happen at any point in the time series of a single lake, or the entire time series could include only one of the two alternative stable states. To accommodate these alternative stable state constellations (for each of which we made a separate simulation scenario, (see Supplementary Note 2, "Simulations of alternative stable state constellations"), we forced the alternative stable state scenario to be constructed of 1/3 of data with one state, 1/3 of data with the second state and 1/3 of data where both alternative states could occur. In the latter case, the alternative stable state appeared after the first 20% but before the last 20% of the time series. Since the variability and range of $x$ (nutrient) and $y$ (chlorophyll -a response) is simulated as close as possible to the real-world data in all scenarios, the measures taken here (variable time series and combination of different alternative stable state scenario constellations) produce a simulation as close to the real-world data as possible. Specifically, we found the real-world nutrient data to be normally distributed, with total nitrogen (TN) having a range between 0.33 and 4.93 mg/L and a constant coefficient of variation (CV, with a mean CV of 0.35) across this range (the same is true for total phosphorus (TP) at a shorter range). Hence, for each simulated lake, the x were generated based on this mean range and CV. Furthermore, the resulting $yi$ and $yj$ were randomised by using a Gamma distribution (using the rgamma function in R). We used a Gamma distribution because the chlorophyll-a concentration also follows a Gamma distribution. The variability of a Gamma distribution is expressed by the *shape* variable. The variability of chlorophyll-a, its *shape* value, equals 2.63. This *shape* value was used in the Gamma distribution of $yi$ and $y$. The final calculated $yi$ and $yj$ had, therefore a random *rate* calculated as *shape/yi* or *shape/y*. Hence, their variability in the y dimension was close to the true chlorophyll-a variability.

For the scenario without alternative stable states, both populations of data had the same intercept and slope:

Population i: $ai = 0$, $bi = 40$
Population j: $aj = 0$, $bj = 40$.

Please see Supplementary Note 2 for further simulations of different potential constellations of alternative states. There we show that our approach finds alternative stable states in response to nutrient concentration, even if they appear in time series from different lakes.

**Assessment of diagnostic tests or proxies of alternative equilibria.** We modelled the response of chlorophyll-a to TP and TN using generalised-linear models[3] with Gamma distribution and an identity link on untransformed data for single-year and multiple-year means up to 5-year means. We used the Gamma distribution, as chlorophyll fit this significantly better than a normal or log-normal distribution. We used $R^2$ of the model along with the patterns of residuals, and finally, we plotted the kernel density of the chlorophyll-a values as diagnostics of the presence, absence or prevalence of alternative equilibria in the simulated and real work data.

The comparison of how the diagnostics/proxies of alternative stable states respond to the variation in the prevalence of alternative equilibria in the simulated datasets provides a robust assessment of their ability to identify both the presence and absence of alternative equilibria. It is the response of these diagnostic tests over time, with the increase in the temporal perspective as more years are added to

the mean values of TP, TN and chlorophyll-a, that are key to the identification of the presence and or absence of alternative equilibria in a given dataset. The simulations show that a dataset which contains alternative equilibria will show (1) no improvement in $R^2$ as the temporal perspective of the data increases (more years in the multi-year mean); (2) an increased bimodality in the residuals of the models of nutrients predicting chlorophyll-a will increase as more years are added to the multi-year mean and (3) the kernel density function of chlorophyll-a will display increasingly bimodality as more years are added to the mean. In the absence of alternative equilibria, the patterns differ with an $R^2$, and increase in unimodality of residuals and a consistent unimodal pattern in the kernel density function. Thus, the diagnostic tests provide a robust test of both the presence and absence of alternative equilibria in a given dataset.

### Alternative stable state assessment for real data with limited data range
It could be the case that alternative stable states do not appear in the full dataset but only in a limited TN and TP concentration range. We filtered and re-analyzed the data, only keeping data points within the following two ranges: - TN concentration = 0.5–2 mg/L – TP concentration = 0.05–0.4 mg/L. In the filtered data, 1329 out of the original 2876 single-year data points, 289 out of 1028 3-year mean data points and 212 out of the 864 five-mean year data points remained. The filtered data consisted of data points from 550, 48 and 27 lakes for the single-year data, 3-year means and 5-year means, respectively. The smaller range resulted in lower $R^2$ of the models, yet the pattern that multi-year means result in higher $R^2$ compared to single-year data was largely consistent, apart from the 5-year mean TN models for which both, the single-year and mean data resulted in very low $R^2$ (Supplementary Fig. 6). Furthermore, due to the lower number of samples, the errors of all proxies are higher, making conclusions more difficult than for the full data. Still, we do not see any clear indication of alternative stable states in the scatter plots (two groups of dots are not appearing (Supplementary Fig. 5), the Kernel density plots (or model residuals (Supplementary Fig. 6)). i.e. no signs of bimodality in residuals or Kernel density plots. Please see details on this analysis in the supplementary material.

Details and the R code for the steps for the random multi-year sampling can be found in the supplementary materials.

### Reporting summary
Further information on research design is available in the Nature Portfolio Reporting Summary linked to this article.

## Data availability
Data were collected from two open-access databases: LAGOSNE database—using the R LAGOSNE package version 2.0.2 and from the Danish ODA database- https://odaforalle.au.dk/main.aspx

## Code availability
All codes are available here: https://git.ufz.de/graeber/alternative-stable-states-do-not-stand-the-test-of-time

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

## Acknowledgements

C.D.S. and T.A.D. would like to thank June and Derek Sayer for extraordinary support over many years. The authors of this work have been supported by a number of projects over the elephantine gestation period of this manuscript. These include support from the Poul Due Jensen Fonden, Danmarks Frie Forskningsfond Natur og Univers project GREENLAKES (No. 9040-00195B) and the UFM-funded project LTER_DK for Long Term Ecosystem Research in Denmark. In addition, support was provided by The European Union's Horizon 2020 research and innovation programmes under grant agreement No 869296—The PONDERFUL Project", TREICLAKE under grant agreement No 951963, and the AQUACOSM project and by the European Commission EU H2020-INFRAIA-project (No. 731065) and AQUACOSM plus (No. 871081). E.J. was also supported by the TÜBITAK outstanding researcher programme 2232 (project 118C250) and AnaEE, Denmark. The work of D.G. was funded by the Fourth Period of Programme-oriented Funding, Helmholtz Association of German Research Centres, Research Field Earth and Environment.

## Author contributions

The study was initially conceived by T.A.D. and C.D.S. with later input from D.G. D.G. and T.A.D. developed the statistical approach and D.G. performed the analysis. E.J., M.S., T.L.L., and L.S.J. curated the Danish lake data. T.A.D. wrote the manuscript with input from co-authors.

## Competing interests

The authors declare no competing interests.
