## [Peer Review File · Nature Communications]

Reviewer comments, first round review –

Reviewer #1 (Remarks to the Author):

Dear Editor,

The ms by Davidson et al explores the validity of the theory of alternative stable states in shallow lakes. Using empirical observations from 902 lakes in Denmark and N America over multiple years they do not find an expected from theory nonlinear relationship between nutrient loading and chlA. In addition, they analyse three lake sediment cores to show that loss of submerged vegetation in the three studied lakes did not happen in an abrupt way but rather smoothly. Based on these two lines of evidence, the authors conclude that the theory of alternative states does not stand the "test of time" in shallow lakes.

I really enjoyed reading this paper. It revolves around a fascinating question that has been causing a lot of discussion lately: whether tipping points and alternative states exist in nature? The authors refer especially to a recent paper (Hillebrand et al 2020) where the matter arises in a rather strong statement of lack of thresholds in ecosystems, and there is the impression that this ms follows partly a similar approach to explore the existence of alternative states in shallow lakes.

Allow me first a rather epistemological comment. The results are based on a large dataset from a wide range of lakes and three detailed paleolimnological records. Even if we accept that the method selected for the analysis is robust (see points below), does the absence of bimodality prove the lack of alternative states in shallow lakes? One could claim that finding a bimodality pattern does not prove that alternative states exist, but the same holds for the opposite statement made by the authors (as well as for the cited Hillebrand paper). Statistically, there will always be cases where a nonlinear model will fit better a linear one: does that prove alternative states? On the other hand, everything is linear when looked on a short limit, or slow time scale. Put differently, it is easier to disprove the existence of alternative states rather to prove them. Actual experiments are the only true testing (Petraitis 2013), and even then evidence is not straightforward (DeRoos et al 2005).

I have few concerns to address that are related to conceptual and methodological issues.

1. It is hard not to expect thresholds in ecological systems as there are limits (physiological, functional, ecological) where responses are other than linear. I would expect similar thresholds to exist in shallow lakes under some conditions. The question then is not if thresholds (and potential bistability) exist, but under which conditions (or if you like how often are these conditions met). Patterns of bimodality, the fit of a linear model, or thresholds in timeseries may be caused by slow responding systems where transients are recorded but not actual equilibrium states (for example Hughes et al 2013 TREE). In the phenomenological analysis as the ones performed by the authors, it is hard to reject this alternative hypothesis. Instead, it is the mechanisms (positive feedbacks) that are needed for alternative states to exist.

Indeed, the authors state in the discussion that their findings do not refute the demonstrated positive feedbacks in shallow lakes. However, in the introduction they question the utility of alternative equilibria (in shallow lakes I assume, l 98) and conclude that alternative equilibria do not help explain the patterns found in lakes. These two statements are contradicting (or at least confusing) and the top-down vs bottom-up explanation is unclear how it connects to them. If the patterns expected do not justify the existence of alternative states, one could also argue that the positive feedbacks are not supported. If instead the authors have an explanation for the lack of patterns (clarifying their discussion of the top-down and bottom-up forces), this could be more useful for understanding dynamics in shallow lakes and when alternative states can be observed, rather than simply rejecting their existence.

Additionally, I would be intrigued to see more discussion in the "utility" of the idea of alternative states "versus the previously held notion of linear gradual response" (l 98-99). In sum, I think the

ms would benefit by a discussion of what the results mean for understanding ecological dynamics (at least in shallow lakes), and for understanding alternative states in general (see the interesting work by Norberg et al Ecol Let 2022).

2. The authors use 3 ways to disprove the existence of alternative states on two different types of data: a spatial dataset of 902 lakes. However, I'm afraid the description of the methods is insufficient to properly understand what the authors did. They provide the code but a detailed description would be helpful.

a, I think I understand what the authors assume but the rationale of the expectation of the R2 and bimodality as a function of the multiyear mean needs to be clearly explained.

b, why was not a lake eutrophication model with alternative states used (like Carpenter et al 1999) as test model?

c, the bootstrapping approach is unclear on what it achieves and why? An example of how it is performed in a record with multiple consecutive years available could help.

d, the data description is a bit limited. Table 1 summarises some characteristics but I fail to see how the data for a specific lake look like and the period they cover. Some suppl info on the spatial distribution and temporal coverage would help.

e, the GLM used has as only covariates TP and TN? Are there any fixed or random factors (like lake ID) defined? What about area as a covariate or depth? Some basic description of the dataset would have been valuable (like correlations between lake properties and chlA). Were there any data normalisation (due to the different levels of chlA across lakes)? Model variance explained is only mentioned but no effect sizes of TP/TN nor significance. Could the authors justify the above choices?

f, how many of these lakes are within the range of expected bistability, or beyond the threshold of supporting macrophytes?

g, was there particular treatment of the lakes that experienced algal blooms? Should there have been to remove potential outliers?

h, how much of multiple testing error is present in the dataset? How independent are all these lakes? I assume there are lakes that are close and there is some degree of spatial correlation?

i, why 5 years is a good scale for lake dynamics? why not longer? did you have enough lakes to try 6 or more years?

3. It is not clear why the authors chose only these particular 3 paleolimnological records. What makes them so representative? Are they connected to the spatial dataset? I see the added value of the paleorecords in addition to the spatial dataset, but it appears rather ad hoc- it is not at the same level in terms of the size of database as we are only talking about three trajectories and it is unclear whether for the same lakes in shorter time scales (as the authors claim) alternative states have been suggested or not.

In addition, I am not convinced of the breaking point found in principal curve in the 3 records (unless I am confused on what it was estimated). Have you tested alternative models (with no breaking point) to conclude the BP model is the best?

4. The other important variable in the alternative states paradigm in shallow lakes is macrophytes. However, this variable is not mentioned in the spatial dataset. It is unclear what is the fate of macrophytes along the nutrient gradients reported or in respect to chlA. Are these data not available? If there are available for a subset of lakes it would be worth examining the response. In theory it is the competition of macrophytes to phytoplankton that determines the alternative states so the relative presence of the two as function of nutrients is of interest.

5. In terms of the analysis on the spatial datasets, my major concern is the assumption that all lakes in the dataset respond in the same bimodal way. In other words, the authors compare the expectations from 2 either/or models: with or without alternative states? But what if (which I think is really plausible) both expectations coexist? This should create a third expectation: let's assume the simplest, 50-50 of both simulated results. How would the expected patterns look like? In fact, alternative states may be less common than 50% so a mix of 70-30 (no alternative- alternative) may be more realistic. I assume this would give patterns close to the ones of no alternative states. Thus, we cannot exclude the hypotheses, that the empirical patterns are due to complete absence of alternative states, or partial presence. In other words, it will be hard to confirm the author's conclusion that alternative states do not stand the test of time.

Putting this main comment aside, few more others.

According to the rationale of the paper, the longer averaged chlA will reveal best the true pattern of potential bistability. Fig 2B shows that the distribution of the significance of the Dip test for the 5 years average follows the expectation of the pattern with alternative states (Fig 2A) even if the mean is insignificant. On the other hand, the GLM model and residuals fit the pattern of no alternative states. What is more important to consider?

The actual density plot of chlA and TN/TP would be helpful to visualise the modality in the data.

Also, the authors state that bistability is expected in the ranges of TP 0.05 - 0.2 mg/L (l 78). Fig 1 shows ranges way beyond these limits. If one would focus on these ranges and redo the analysis, what would be the result? Actually it is noteworthy that for values above 0.2 chlA is rather low (or at least lower than for the range 0.05 - 0.2). How is this explained? Does this already suggest that these points are not relevant for the question at hand, or even that they dispute the whole argument of a turbid state due to eutrophication?

In addition to that, is the expectation to have alternative states along a gradient of TN or TP? Your model suggests TP is better descriptor than the TN and the interaction model. If so, why are both analysed? Have you considered a 2D density plot of chlA of the gradients of TN and TP?

As stated above, I think I understand the rationale of the linear model test. Nonetheless, have you tried to fit nonlinear GAMs to compare across models for the existence of nonlinear effects? From fig1A, the linear model seems not the best one to fit at least for single year data.

Could you define the levels of chlA that a lake should be considered as turbid (and thus belonging to the turbid state)? My upfront apology for oversimplifying, but one should expect to see a clear figure of the raw data (as in Fig1 single years) where the expectation of what is the two alternative states are shown.

How does the smaller subset of lakes affect the results? With a 5 year mean there are 99 lakes of the total 902 analysed to infer the pattern. Could it be that this subset of lakes are particular in some respect? Does the result for this subset of the single year is analogous to any random subset of 99 lakes from the rest?

Other comments:

l 65: resistance to env change, i think you mean stable

l 77: stabilised by positive feedbacks is paradoxical as positive feedbacks are destabilising

l 82-85: this part needs to be expanded to be understandable and also connected to the results of the paper

l 94-95: alternative states do not necessarily lead to abrupt changes in temporal dynamics (see Hughes et al 2013 TREE)

l 120-121: why is it notable? and what does it mean?

l 117-136: there is some repetition on the R2 between the two paragraphs

l 139: what is the multi-indicator?

l 140: why only the 3 particular records?

l 186: significant linear? can you specify?

l 187: what does moot mean?

l 191: persistent drying maintains persistent or transient dynamics? clarify

l 209-210: related to my point 1 above: gradual change over time does not exclude the existence of thresholds

l 222-223: does the data presented here confirm the greater plant abundance variation?

Reviewer #2 (Remarks to the Author):

Davidson et al. use two datasets to test empirically for the existence of alternative stable states in shallow lakes, one a contemporary dataset comprising 902 lakes where they explored relationships between nutrient and chlorophyll a concentrations in the water column, and the other a

palaeolimnological analysis of biological communities in three lakes. There is much to like in this paper. The idea being tested is important with impactful implications. There are, however, some issues with the study that mean that the findings are just not sufficiently compelling to convince the reader of their robustness or generality. I think that the authors are on the right track with their efforts, and I hope that they find my comments and recommendations below useful in strengthening this interesting and exciting work.

The contemporary analyses:

The authors utilise a large dataset comprising 902 lakes from North America and Denmark to test for multimodality in water column chlorophyll a concentrations at this large spatial scale and to explore the strength and nature of relationships between water column nutrient (total P and total N) concentrations and chlorophyll a concentrations. This all seems perfectly reasonable on the face of it, but I think there are a number of things that need additional thought:

- The HD test was used to test for multimodality in the ecological response variable, but why not also use it to test for multimodality in the pressure-ecological response relationship (that is, on the residuals of the nutrient-chl linear model), given that such a bimodal response is predicted for alternative stable states as per the conceptual model depicted on Fig 1?
- The HD test for the 5 year mean was very close to significance (looks like $0.05 < p < 0.1$ on Fig 3), but this actual p-value was not reported in the text and the important implications of this were not discussed. Surely this is a critically important finding in the context of the manuscript? Using mean data averaged over the longer term clearly increases strongly the likelihood of finding bimodality in the data, yet this important finding is just brushed over uncritically. If the authors want to address the stated research questions in a robust and convincing manner, this important finding needs to be addressed head-on in the manuscript and not brushed under the carpet as it does not fit with the narrative of the manuscript, as seems to the reader to have been done here.
- I have a fundamental issue with the a priori assumption of the conceptual model and simulated data that alternative stable states are to be expected over the entire pressure gradient (in this case, water column nutrient concentrations). Why would we expect similarly strong bimodality in the chl a response at very low and very high nutrient concentrations, as depicted in Figs 1 and 2? I argue that this is in fact not the expectation, and rather that we should instead expect bifurcation in the response above a certain "threshold" on the pressure gradient. In fact, this is exactly what seems to be occurring on the scatterplots of the real-world empirical data shown on Fig 2. Testing explicitly for changes (increases) in the variance of the data over the pressure gradient (analogous to that done by Hillebrand et al., reference [14] in the ms) would address this. My strong recommendation here is that the authors revisit their a priori expectations, simulations and analyses to develop a much more robust and convincing set of tests of their research questions.
- Please note that the analyses need to be reported in the results in the same order as the research questions were posed. This is not the case at present and is confusing for the reader.
- Finally, the single year vs multiple year analysis of this dataset is not actually relevant in terms of the hypotheses. I recommend relegating that research question and set of analyses to the supporting information. It adds unnecessary and tangential detail to the manuscript that undermines the central message and reduces the impact of the work.

The palaeolimnological analyses:

While I very much appreciate the inclusion of the long-term perspective the inclusion of these data provide, the scale of the analysis in terms of the ability to provide a general overview of ecological responses to eutrophication is lacking, and is very inconsistent with the very broad and general overview provided by the spatial analysis of the contemporary data. Showing detailed responses of just three lakes is just not sufficient to indicate a general lake response. Unfortunately, inclusion of data from many more lakes is really needed to achieve this.

Reviewer #3 (Remarks to the Author):

Nat Comm

My focus is on the paleolimnological data because the modelling is not within my scope of research.

Lake sediments are a great way to assess catastrophic regime shifts because the indicator taxa (in this case Cladocera) should also change abruptly when the habitat shifts between a clear and macrophyte dominated to turbid and phytoplankton dominated system. In this study, lakes that have been historically documented to shift from macrophyte to phytoplankton dominated systems make ideal test sites. I will say that Felbrigg Lake seems an odd choice to include as it has never completely transitioned between so-called alternative states (it appears to currently be in the process of losing its macrophyte cover). The other two study sites have a dashed red line denoting "plant loss", whereas in Felbrigg a similar line denotes "loss of zooplanktivore". This may lead to confusion among readers. Overall, the paleolimnological data show that the Cladoceran indicators show gradual, rather than rapid, changes preceding and following a change from clear to turbid conditions. These data help strengthen conclusions reached from the modelling.

Minor comments:

Line 41. Add comma after "Since its inception"

Line 103. "2986 lakes years" Seems like odd terminology?

Line 142. "Prior to the modern era and the major development of eutrophication in the three lakes the plant community was dominated by less nutrient tolerant submerged plants including charophytes (all lakes)"

Not really accurate for Lake Sobygaard - only one peak in charophyte oospores at ~83 cm

Line 232. It is noted that fish kills have been recorded in the sediment record at Felbrigg; however, it is not clear what proxy(ies) indicates the loss of fish.

Line 233. "The change in macrophyte composition in the sediment cores in all three lakes mirrors those observed as a result of nutrient enrichment over time..."

Ok, but this is hard for readers to assess as little info is given on when nutrient enrichment has occurred over time. No mention is given on TP values over time in the 3 lakes. Likewise, on the sediment profile diagrams there are no proxies of lake production (e.g., fossil pigments, TOC).

Line 237. "In addition, in the palaeolimnological records analysed here, pelagic taxa dominated the assemblages present prior to the complete total loss of submerged vegetation."

This is true, but in Felbrigg Chydorus dominates much of record but declines towards surface. In Sobygaard, benthic taxa only ever made up a very small portion

Perhaps a good point to stress here is that in all cases with plant loss there was no rapid regime shift in Cladoceran assemblages. In Sobygaard, planktonic Bosmina remained dominant, and in the other two study sites there was a decline in benthic Chydorus and an increase in planktonic Bosmina, but nothing that would constitute a regime shift. Thus, what is observed is gradual, rather than step-changes.

Table 1. Better to report TP in more familiar units of $\mu\text{g/L}$, rather than mg/L ?

Response to reviewers

We would like to thank the reviewers for their general enthusiasm for the work and for their insightful, and at times challenging, critique of the study. These comments and questions have stimulated us to add, and/or change, a number of elements of the study. This has extended the statistical analysis to address the reviewers' comments and we give an overview of the main changes here and more detailed responses to specific points below.

First, we would like to address the general point on the theme of proving or disproving the existence or perhaps if we put it in a less polemic framework, the prevalence or rarity, of alternative equilibria in the data we use here and then shallow lakes generally. As the prevailing view in the literature is that the theory is well-founded for shallow lakes the evidence required to question or disprove the theory is arguably greater than that needed to maintain the consensus, despite the scant empirical evidence of stable alternative equilibria^{1,2}. To put it more clearly, just showing the absence of patterns that reflect alternative states is not sufficient, we need to demonstrate the opposite pattern convincingly. We have tried to approach this work in a balanced way and if the data here showed the presence of alternative states in the diagnostic tests we would be presenting those data. What we seek to do here is use the data simulations and a large data set, along with the long-term perspective of palaeolimnology to make a balanced assessment of the presence, absence and with the new scenarios, prevalence of alternative equilibria in shallow lakes.

We assert that the results are very clear and one way of illustrating this is to turn the focus of the paper on its head and imagine we are back in 1993 and trying to demonstrate the existence of this very new theory of alternative equilibria in shallow lakes using empirical time series data. Using our approach, we find no evidence for the existence of alternative equilibria at any of the temporal perspectives we use in the study. The only possible evidence for alternative equilibria would be equating the lack of a strong linear relationship between nutrients and chlorophyll-a at the single year scale (the cloud of data) with the existence of alternative equilibria. However, this is a conjunction fallacy³, which is the inference, in violation of the laws of probability, that a conjoined set of two or more conclusions is likelier than any single member of that same set. In this case, the conclusion of the absence of a strong relationship at sub-annual or annual scale is joined to the conclusion that alternative-stable states or equilibria are present. This conclusion would require further assumptions, specifically that bi-modality of the response (which is not evident for Chlorophyll-a at annual scale) is a result of underlying complex feedback loops. However, even if bi-modality would exist in our annual data, it would not necessarily be a result of alternative stable states due to biological feedback (please also see the detailed assessment of long-term data from single lakes in the SI which examine this in more detail). In contrast, we propose that the annual variability in the response of lake phytoplankton to nutrient inputs is merely stochastic. In contrast, the long-term patterns of change from the palaeo record combined with the comparison of real world data with the simulations at a range of temporal scales evince a longer-term, deterministic, linear relationship between nutrient enrichment and phytoplankton biomass in shallow lakes. Which is a simpler explanation than alternative stable states for the real world patterns we see in lakes in response to nutrient enrichment.

1. A major comment by the reviewers can be summarized as 'it was difficult to follow the statistical approach, randomized sampling and bootstrapping procedures'. We have amalgamated all the methods into a single methods section, rather than spread across three documents. In addition,

we have added a more extended description of the randomized resampling procedures and bootstrapping approach, including a flow diagram of the data preparation, given in the methods section.

2. A second area of concern was the way the simulations were conducted and also on the ability of the statistical approach to tests or assess the presence of absence of alternative equilibria in empirical data. As a result, we revised the general approach to the simulations to be as similar as possible to the real dataset. Then we increased the number of simulation types, to address reviewer 1 point 5 (what if only a proportion of lakes display alternative states?) so as to better assess the ability of the different proxies to detect alternative stable states and different constellations of alternative equilibria. Within the simulations:

- i) We adjusted the number of lakes in the simulated dataset to the true number of lakes found in the five-year means, hence, we simulated a dataset with 99 lakes.
- ii) We found the real nutrient data to be normally distributed, with total nitrogen (TN) concentrations having a range between 0.33 and 4.93 mg / L and a constant coefficient of variation (CV , with a mean of 0.35) across this range (the same is true for total phosphorus (TP) concentrations at a shorter range). Hence, we constructed x (the nutrient data) based on a normal distribution based on the range of total nitrogen and with a standard deviation of $x * CV$.
- iii) The number of years from which means could be calculated was highly variable between the lakes in our datasets. We found between 5 and 20 years of data for the real data for which five-year means could be constructed. Hence, we randomly assigned between 5 and 20 years of data for each lake in the simulated dataset.
- iv) The variability of the simulated response of y (simulating Chl-a) follows a Gamma distribution (as does Chl-a), with a variability (*shape* variable) of 2.63, which equals the variability of the real world Chl-a data.
- v) We improved the simulation of the scenario where alternative stable states are present in the data. The manifestation of alternative stable states in the simulated or real world data could happen at any point in the time series of a single lake, or the entire time series could include only one of two alternative stable states (Therefore, the alternative stable state scenario of the revised simulation was constructed of $\frac{1}{3}$ of data with one state, $\frac{1}{3}$ of data with the second state, and $\frac{1}{3}$ of data with both alternative states, where the alternative stable state appeared after the first 20% but before the last 20% of the time series.
- vi) We furthermore expanded the range of scenarios to show what happens for a range of alternative state constellations (see supplementary material): 1) No alternative states 2) Occasional alternative states in a single lake due to high random inter-annual variability (which we dubbed alternative unstable states). 3) Alternative-stable states in the same time series, which is closest to the simulation done in the original submission 4). Alternative stable states present in different lakes. This last scenario was chosen since we mostly had short time series. Here, the probability of two alternative stable states appearing in the same time series is low. Hence, we also assessed what happens if only one or another alternative-stable state occurs in an individual time series. With these simulation scenarios, we show that the approach is capable of finding alternative stable states within one time series but also if they would only occur within different time series. We furthermore find it to be robust against high random variability within a time series, which could falsely be assumed to resemble tipping-point behavior or alternative stable states.

3. The points raised by the reviewers made us rethink the use of the Hartigan’s dip test (HD test, Hartigan and Hartigan 1985) and we reappraised its use after the recommendation of reviewer 2. The HD test aims to show existence of unimodal versus multimodal distributions, where a significant p value (usually < 0.05) indicates multimodality. We tested this for two simulated datasets with and without alternative stable states (see section “simulation approach in new methods section” for details on the construction of the simulated datasets). These simulations were improved after comments from the reviewers to better resemble the real world data. With the revised simulations, we found the HD test to be highly sensitive to small scale, low bandwidth variability of unimodal distributions for multi-year means. Even if the data came from a simple simulated linear relationship between x and y , p values were still lower with more mean years, indicating a higher probability of multimodality (Fig. 1, bottom right panel). Furthermore, the p values of the HD tests failed to indicate alternative stable states even for simulated data which included these (Fig. 1, top right panel). In contrast, high bandwidth kernel density plots reveal small secondary peaks for the dataset with alternative stable states, which is not the case for the data without alternative stable states (Fig. 1). Furthermore, the low bandwidth lines (Fig. 1) indicate random appearance of small peaks for data with three and five mean years, which mislead the HD test. In contrast to the HD test, the Kernel density plots in combination with residual distributions and R^2 of the generalized linear models clearly indicate the presence or absence of the alternative stable states in the simulations (Fig. 2 below equalling Fig. 3 of the main text). This explanation is not included in the text of the resubmitted paper, as the HD analysis no longer appears in the paper.

Fig. 1 Kernel density plots with low and high bandwidth compared to p values of HD tests for the two simulations also shown in the main text. Distributions are based on 1000 iterations of the hierarchical bootstrap procedure.

See the main text for the residual and R^2 distributions of the simulated datasets (Fig. 3 within the main text) and see sections “General approach” plus “Real-life simulation scenario” for details on the construction of the simulated datasets).

REVIEWER COMMENTS

Reviewer #1 (Remarks to the Author):

Dear Editor,

The ms by Davidson et al explores the validity of the theory of alternative stable states in shallow lakes. Using empirical observations from 902 lakes in Denmark and N America over multiple years they do not find an expected from theory nonlinear relationship between nutrient loading and chlA. In addition, they analyse three lake sediment cores to show that loss of submerged vegetation in the three studied lakes did not happen in an abrupt way but rather smoothly. Based on these two lines of evidence, the authors conclude that the theory of alternative states does not stand the "test of time" in shallow lakes.

I really enjoyed reading this paper. It revolves around a fascinating question that has been causing a lot of discussion lately: whether tipping points and alternative states exist in nature? The authors refer especially to a recent paper (Hillebrand et al 2020) where the matter arises in a rather strong statement of lack of thresholds in ecosystems, and there is the impression that this ms follows partly a similar approach to explore the existence of alternative states in shallow lakes.

Allow me first a rather epistemological comment. The results are based on a large dataset from a wide range of lakes and three detailed paleolimnological records. Even if we accept that the method selected for the analysis is robust (see points below), does the absence of bimodality prove the lack of alternative states in shallow lakes? One could claim that finding a bimodality pattern does not prove that alternative states exist, but the same holds for the opposite statement made by the authors (as well as for the cited Hillebrand paper).

Reply: We would like to thank the reviewer for the positive comments on the paper and for the enthusiasm for the work. The question of providing evidence that absolutely proves existence or non-existence of a phenomenon such as alternative equilibria is the kernel of this paper. If we take the view that finding bimodality does not prove alternative equilibria and that the same holds true for the opposite we end up with the conclusion that no empirical evidence can ever prove or disprove any ecological theory. We would politely disagree with this strong statement, it is very true, however, that it is hard to prove the absence of bimodality in single, or subset of lakes. In our revised paper we sought to establish framework using the different simulation scenarios where we could provide diagnostic indicators providing support for either the presence or absence of alternative equilibria in a given dataset. To this end, we expanded the simulation scenarios here in order to be better able to assess the ability of the diagnostic indicators we use to test for the existence of alternative equilibria (see also the response above to the editor)

Statistically, there will always be cases where a nonlinear model will fit better a linear one: does that prove alternative states? On the other hand, everything is linear when looked on a short limit, or slow time scale.

Reply: It is true that there will always be cases where nonlinear models fit better when things are random, but the patterns we are investigating are not random. The study does not focus on the efficacy of different types of models, but rather uses a single modelling approach (GLM) to compare simulated with real world data across a range of temporal scales. The method relies on the repeated use of the same type of model and addressing the central question would, in our opinion, not benefit from trying the fit non-linear models. The idea that everything is linear if a short part of a gradient is selected or a short time scale is also true but here we looked at the relationship between nutrients and chlorophyll at a range of timescales and on long gradients of nutrients, also on shorter gradients, in simulated and real world data. The simulated data are useful as we can see how data containing alternative equilibria would look at short and longer timescales and compare them with real world data. But the reviewer is right – it is very hard to disprove or prove the theory with a single shot – and when the scale of the processes driving the patterns operate on multi-year scales. That is why we combined data simulations, real world contemporary data and crucially the palaeo data, to combine and accumulate evidence from different approaches and timescales.

Put differently, it is easier to disprove the existence of alternative states rather to prove them.:

Reply: To better address this interesting point raised by the reviewer we expanded the data simulation element of the study to better test both for the existence or absence of alternative equilibria in real world data. See general explanation in the new method section. The simulated data present the patterns expected in the diagnostic indicators we use to assess the existence or lack of alternative equilibria in the data at difference temporal perspective, i.e. R^2 of the model, patterns in the residuals and density plots (replacing the HD dip test – see comments above). We think that the combination of the methods used here, the different simulations and data sets provide a means to assess whether the real world data have patterns reflecting the prevalence of alternative states or in contrast a linear response, or even something in between, it is an objective test. If we took only single year data the results are less clear, which in part explains the plausibility of the idea of alternative equilibria, but the multiple year data provide a clear test which has no *a priori* likelihood of proving or disproving the existence of alternative states. In conclusion the reviewer is correct – it is hard to prove the absence of bimodality in a single, or subset of lakes – but as stated above the data presented here show the theory has, at best, very limited applicability in real world shallow lakes when a long-term (5 years or longer) perspective is used.

Actual experiments are the only true testing (Petraitis 2013), and even then evidence is not straightforward (DeRoos et al 2005).

Reply: The reviewer is correct, experiments are extremely useful in delving into the processes determining the observed patterns. However, experiments are always limited in space, time and replication. Furthermore, our findings point to a multi-year scale in the processes shaping the nutrient to chlorophyll-a relationship in shallow lakes and experiments are not well suited to studying such long timescales. We assert that only empirical evidence can help us survey whether ideas on ecological theories hold true at relevant ecosystem scale over multiple lakes.

I have few concerns to address that are related to conceptual and methodological issues.

1. It is hard not to expect thresholds in ecological systems as there are limits (physiological, functional, ecological) where responses are other than linear. I would expect similar thresholds to exist in shallow lakes under some conditions. The question then is not if thresholds (and potential bistability) exist, but under which conditions (or if you like how often are these conditions met). Patterns of bimodality, the fit of a linear model, or thresholds in timeseries may be caused by slow responding systems where transients are recorded but not actual equilibrium states (for example Hughes et al 2013 TREE). In the phenomenological analysis as the ones performed by the authors, it is hard to reject this alternative hypothesis. Instead, it is the mechanisms (positive feedbacks) that are needed for alternative states to exist.

Indeed, the authors state in the discussion that their findings do not refute the demonstrated positive feedbacks in shallow lakes. However, in the introduction they question the utility of alternative equilibria (in shallow lakes I assume, l 98) and conclude that alternative equilibria do not help explain the patterns found in lakes. These two statements are contradicting (or at least confusing) and the top-down vs bottom-up explanation is unclear how it connects to them.

If the patterns expected do not justify the existence of alternative states, one could also argue that the positive feedbacks are not supported. If instead the authors have an explanation for the lack of patterns (clarifying their discussion of the top-down and bottom-up forces), this could be more useful for understanding dynamics in shallow lakes and when alternative states can be observed, rather than simply rejecting their existence.

Reply: We thank the reviewer for highlighting the insufficient discussion of this apparent contradiction. We have included further text and references in an attempt to explain the idea that it is the instability between years in these positive feedbacks, that is of relevance here. As stated, the trophic interactions and biogeochemical processes favoured in the presence of plants do alter conditions to further favour plants. However, these processes appear unstable and whilst they can suppress chlorophyll a in the face of higher nutrients in the short term (1-3 years), the data presented here demonstrates that at the longer time scale the bottom-up forces prevail and TP and TN determine chlorophyll-a concentration, with a very high degree of predictability. It is entirely due to these trophic interactions or feedbacks that the single year perspective provides such an equivocal picture on the presence or rarity/absence of alternative equilibria.

Additionally, I would be intrigued to see more discussion in the "utility" of the idea of alternative states "versus the previously held notion of linear gradual response" (l 98-99). In sum, I think the ms would benefit by a discussion of what the results mean for understanding ecological dynamics (at least in shallow lakes), and for understanding alternative states in general (see the interesting work by Norberg et al Ecol Let 2022).

Reply: We thank the reviewer for the interesting and constructive suggestion. The Norberg et al paper is very insightful look at how scientific disagreement often does not make sense in hindsight. We have added more discussion of the implications of the findings here, in applied lake management terms and also more broadly in the need for a longer term perspective in ecology and ecological theories.

We have also tried to avoid the more polemic language from the previous version in the spirit of the Norberg et al paper and added some text at the end of the discussion on both wider and more applied implications.

2. The authors use 3 ways to disprove the existence of alternative states on two different types of data: a spatial dataset of 902 lakes. However, I'm afraid the description of the methods is insufficient to properly understand what the authors did. They provide the code but a detailed description would be helpful.

a, I think I understand what the authors assume but the rationale of the expectation of the R² and bimodality as a function of the multiyear mean needs to be clearly explained.

Reply: We have expanded the explanation of the numerical approaches and the bootstrapping procedures and include a flow diagram in the methods to show more simply the procedure. We set out more clearly how the multi-year means were used and how we use the simulated data to derive expected responses in R², residual patterns and the kernel density of chlorophyll-a data (alternative to HD test explained elsewhere).

b, why was not a lake eutrophication model with alternative states used (like Carpenter et al 1999) as test model?

Reply: It was not explicitly included but is implicit in our analysis. Carpenter et al 1999 includes two attractors, one for oligotrophy and one for eutrophy. Our method would find these attractors by a combination of low R² and split residuals, the simulations with alternative states included have this pattern.

c, the bootstrapping approach is unclear on what it achieves and why? An example of how it is performed in a record with multiple consecutive years available could help.

We have included a flow diagram to explain the procedure

d, the data description is a bit limited. Table 1 summarises some characteristics but I fail to see how the data for a specific lake look like and the period they cover. Some supplementary info on the spatial distribution and temporal coverage would help.

e, the GLM used has as only covariates TP and TN? Are there any fixed or random factors (like lake ID) defined? What about area as a covariate or depth? Some basic description of the dataset would have been valuable (like correlations between lake properties and chlA). Were there any data normalisation (due to the different levels of chlA across lakes)? Model variance explained is only mentioned but no effect sizes of TP/TN nor significance. Could the authors justify the above choices?

Reply: We have included maps of the spatial distribution of the sites in the supplementary materials. The GLM with the gamma distribution and with TP and TN do a good job of explaining the variation in the chlorophyll-a data. The randomized sampling and bootstrap approach means that there is no need to a random effect of lake ID as each model includes only one observation (though this can be the mean of

several years, per lake. There were no data normalisations carried out and as the data distribution (gamma) was included in the GLM. A previous study on the Danish data used here ⁴ used boosted regression trees to and showed that depth and area had marginal effects on chlorophyll-a.

f, how many of these lakes are within the range of expected bistability, or beyond the threshold of supporting macrophytes?

Reply: A large proportion of the data fall between the range of 40 and 200 $\mu\text{g/l}$ for TP (57% of the data) and 0.5-2 mg/l (67% of the data) N which could be defined as that where bistability is expected

However, as the issue of the limited range of expected bistability was also a point raised by reviewer 2. We addressed this issue by filtering and re-analyzing the data, only keeping data points within the following two ranges: - TN concentration = 0.5 - 2 mg / L - TP concentration = 0.05 - 0.4 mg / L

In the filtered data, 1329 out of the original 2876 single-year data points, 289 out of 1028 three-year mean data points and 212 out of the 864 five-mean year data points remained. The filtered data consisted of data points from 550, 48 and 27 lakes for the single-year data, three-year means and five-year means, respectively. The smaller range resulted in lower R^2 of the models, yet the pattern that multi-year means result in higher R^2 compared to single-year data was largely consistent, apart from the five-year mean TN models for which both, the single-year and mean data resulted in very low R^2 (SFig. 7). Furthermore, due to the lower number of samples, the errors of all proxies are higher, making conclusions more difficult than for the full data. Still, we do not see any clear indication of alternative stable states in the scatter plots (two clusters of sites are not evident, SFig. 6), the Kernel density plots or model residuals (SFig. 7) (no signs of bimodality in residuals or Kernel density plots).

It is notable that the analysis presented here with the multi-year means has remarkable stationarity and predictability in the relationship between nutrients and chlorophyll-a across a very large range of nutrient concentrations, with no evidence of threshold response when a longer-term perspective is taken. It is striking and shows a highly predictable relationship between nutrients and chl-a irrespective of the point on the gradient

g, was there particular treatment of the lakes that experienced algal blooms? Should there have been to remove potential outliers?

None. There were no sites removed as the bootstrapping approach is very robust to outliers and reveals an increased error of the statistical estimates (e.g. R^2) due to outliers. Here, we find our estimates robust (hence low variability of e.g. R^2), especially for more mean years.

h, how much of multiple testing error is present in the dataset? How independent are all these lakes? I assume there are lakes that are close and there is some degree of spatial correlation?

Reply: We assume that the reviewer means increased risk of inflated likelihood to find the alternative hypothesis (type II error / alpha error) by multiple testing? We have no risk of alpha error since we (i) refrain from using p values within correlations and (ii) since we do not do multiple testing but use bootstrap to assess the error of the calculated statistical variables. R^2 is not affected by multiple testing. The lakes are independent entities and are spread over a relatively large area (see maps in methods

section). We do not see a logical dependence here. Only indirect effects via nutrient export or algal export are possible and that would be captured in our correlations.

i, why 5 years is a good scale for lake dynamics? why not longer? did you have enough lakes to try 6 or more years?

Reply: We tried up to seven years, but found a strong increase in bootstrap error of our statistical estimates with more than 5 years due to a lower number of lakes remaining in the dataset. This also shows the usefulness of the bootstrap method and its error estimate to support our findings, as the bootstrap error of all estimates (therefore the width of their distribution) was well constrained for 1, 3, and 5 years.

Methods adapted – flow diagram of analysis added

3. It is not clear why the authors chose only these particular 3 paleolimnological records. What makes them so representative? Are they connected to the spatial dataset? I see the added value of the paleorecords in addition to the spatial dataset, but it appears rather ad hoc- it is not at the same level in terms of the size of database as we are only talking about three trajectories and it is unclear whether for the same lakes in shorter time scales (as the authors claim) alternative states have been suggested or not.

In addition, I am not convinced of the breaking point found in principal curve in the 3 records (unless I am confused on what it was estimated). Have you tested alternative models (with no breaking point) to conclude the BP model is the best?

Reply: We acknowledge that the added value of the paleo data does not and cannot have the same spatial scale in terms of number of lakes. However, their value should not be underestimated as they provide a unique insight into long-term dynamics in lakes and the high resolution cladoceran record provides an integrated picture of ecosystem change –as community composition is influenced by chlorophyll-a, late summer plant cover and zooplanktivorous fish abundance. They are representative of shallow lakes that have lost or are close to losing their submerged plants, with a very representative, eutrophication driven pattern of change in plant species occurrence. The breakpoint model is a regression tree which identifies the point of greatest change in the cladoceran assemblage across the several 100 years of the record. If the catastrophic regime shift occurred in the record with a rapid reorganization of all levels of the food web the largest change in cladoceran community should be at that point. The BP analysis finds the point of greatest difference on the two resultant groups, it is not a rate of change analysis

4. The other important variable in the alternative states paradigm in shallow lakes is macrophytes. However, this variable is not mentioned in the spatial dataset. It is unclear what is the fate of macrophytes along the nutrient gradients reported or in respect to chlA. Are these data not available? If there are available for a subset of lakes it would be worth examining the response. In theory it is the competition of macrophytes to phytoplankton that determines the alternative states so the relative presence of the two as function of nutrients is of interest.

Reply: This is an excellent point, unfortunately macrophyte data are either not available (US sites), or not available at the same resolution (2-3 years apart) as the chlorophyll data. In Denmark they are available for a subset of sites and for a subset of years so cannot be analysed in the same way. However, chlorophyll-a, and its strong correlate Secchi depth, are the two best predictors of plant cover in these same lakes at the single year grain⁴. In general terms, when we are discussing chlorophyll-a we are also by association discussing plants cover (this may not be true for the very low nutrient conditions), as there is a strong negative relationship between chlorophyll-a and plant cover. So, whilst the absence of macrophytes cover data from the paper is a shame it does not detract from the findings of the paper as the existence of each stable state or a gradient response is well-characterized by chlorophyll-a data.

5. In terms of the analysis on the spatial datasets, my major concern is the assumption that all lakes in the dataset respond in the same bimodal way. In other words, the authors compare the expectations from 2 either/or models: with or without alternative states? But what if (which I think is really plausible) both expectations coexist? This should create a third expectation: lets assume the simplest, 50-50 of both simulated results. How would the expected patterns look like? In fact, alternative states may be less common than 50% so a mix of 70-30 (no alternative- alternative) may be more realistic. I assume this would give patterns close to the ones of no alternative states. Thus, we cannot exclude the hypotheses, that the empirical patterns are due to complete absence of alternative states, or partial presence. In other words, it will be hard to confirm the author;s conclusion that alternative states do not stand the test of time.

Reply: Addressed in the new simulation framework.

This is an interesting idea that alternative stable states are present sometimes but not others. Or present in some lakes but not in others. However, the benefit of the use of simulated data means that we could address these concerns by changing our simulation approach to include scenarios where different amount of instability between states occurring within a single site and among sites. We constructed a number of different scenarios to assess the effects of alternative stable states on our proxies and to explain our approach. We based the simulated data on the true range of real nutrient data, variability of the Chl-a response and for 99 lakes (five-year mean data consists of 99 lakes, see also section "General approach"). However, we chose a constant number of years per lake (20 years per lake, which equals the maximum time series length of the real data) to better represent the different alternative-stable state scenarios. We used the following scenarios (see supplementary materials)

1. No alternative stable states (SFig.3, left panels)
2. Unstable alternative states appearing randomly in a time series, therefore no longer-term persistence (stability) of different states exists (Fig. 3, right panels).
3. Alternative stable states appearing within a time series of one lake at any point in time after the first twenty percent of the simulated years and before the last twenty percent of the simulated years (SFig. 4, left panels). The 20% criterion prevents that this scenario partially consists of only five-year means of one alternative state, a constellation which is covered by scenario 4.
4. Alternative stable states only appearing in different lakes, never in the same lake (SFig. 4, right panels).

According to the rationale of the paper, the longer averaged chlA will reveal best the true pattern of potential bistability. Fig 2B shows that the distribution of the significance of the Dip test for the 5 years average follows the expectation of the pattern with alternative states (Fig 2A) even if the mean is insignificant. On the other hand, the GLM model and residuals fit the pattern of no alternative states.

What is more important to consider?

The actual density plot of chlA and TN/TP would be helpful to visualise the modality in the data.

Reply: We have followed the reviewer's suggestion here as upon further consideration the use of the HD test was not optimal here (see above) and instead we used kernel density plots of chlorophyll-a as a diagnostic contributing to the assessment of the presence or absence of alternative equilibria.

Also, the authors state that bistability is expected in the ranges of TP 0.05 - 0.2 mg/L (l 78). Fig 1 shows ranges way beyond these limits. If one would focus on these ranges and redo the analysis, what would be the result? Actually it is noteworthy that for values above 0.2 chlA is rather low (or at least lower than for the range 0.05 - 0.2). How is this explained? Does this already suggest that these points are not relevant for the question at hand, or even that they dispute the whole argument of a turbid state due to eutrophication?

Reply: Analysis on a more limited range was done and the results are presented in the SI. The results show the same pattern as the larger dataset in the residuals and for the R^2 for the combination of TP+TN and for TP, but the pattern is not the same for the R^2 response of TN alone. This is because within this limited range nutrients TP exerts the greatest influence over chlorophyll-a, which was clearly shown by Søndergaard et al (2021), whereas the influence of TN continues over the length the gradient. The smaller number of samples adds some noise to the results but the same patterns in the R^2 and model residuals are present

In addition to that, is the expectation to have alternative states along a gradient of TN or TP? Your model suggests TP is better descriptor than the TN and the interaction model. If so, why are both analysed? Have you considered a 2D density plot of chlA of the gradients of TN and TP?

Reply: There is a great deal of focus on TP in the literature of alternative states and this in part can lead to some suggestion, though no hard evidence, of alternative states as the relationship between TP and chlorophyll-a varies along the TP gradient with other factors, including N. See new section in supplementary information and also reply to reviewer 2. As a result we think it important to include both nutrients in the analysis to demonstrate that both can be important and also their ratio is important in determining the relationship between each respective nutrient and chlorophyll-a.

As stated above, I think I understand the rationale of the linear model test. Nonetheless, have you tried to fit nonlinear GAMs to compare across models for the existence of nonlinear effects? From fig1A, the linear model seems not the best one to fit at least for single year data.

Reply: We did not try to fit a non-linear model as the design of the study was to compare the same modelling approach to all the permutations of simulated and real world data. If there was a better non-linear fit to the data in Fig1A it would not affect the findings of the study.

Could you define the levels of chlA that a lake should be considered as turbid (and thus belonging to the turbid state)? My upfront apology for oversimplifying, but one should expect to see a clear figure of the raw data (as in Fig1 single years) where the expectation of what is the two alternative states are shown.

Reply: As the reviewer acknowledges this is a simplification and the main thrust of the paper here is that there is no critical value that can be defined as 'turbid' as there is no stable turbid or clear water state. Rather nutrient enrichment increases turbidity in a linear predictable way, but this is only evident when the year to year variation due to top down effects, weather patterns and trophic interactions are evened out by a longer-term mean. Part of the problem of trying to fit thresholds, tipping points, points of critical transition, to define the different states is that in empirical data we always find a gradient, there are not basins of attraction around a particular macrophyte abundance that ensures 'clear water' in the longer term. In short the findings here suggest that there is no rationale for classifying a lake as turbid or clear, as there is a gradient response to turbidity.

How does the smaller subset of lakes affect the results? With a 5 year mean there are 99 lakes of the total 902 analysed to infer the pattern. Could it be that this subset of lakes are particular in some respect? Does the result for this subset of the single year is analogous to any random subset of 99 lakes from the

We have altered the simulations to take account of the possible differences in simulations based on a different number of lakes than is present in the real world analysis. Furthermore, we only compare 5-year means and single year data from the same set of lakes (hence for 5 years we plot the single year data and the 5-year mean data in the same line of the plot in Figure 3 and show scatter plots based on the same data). Finally, the bootstrapping approach would show us a high error of our proxies due to an anomalous subgroup of lakes.

Other comments:

I 65: resistance to env change, i think you mean stable

Changed wording

I 77: stabilised by positive feedbacks is paradoxical as positive feedbacks are destabilizing

Changed wording

I 82-85: this part needs to be expanded to be understandable and also connected to the results of the paper

Text expanded to examine this section in more details

*

I 94-95: alternative states do not necessarily lead to abrupt changes in temporal dynamics (see Hughes et al 2013 TREE)

. Reply: Reference inserted and text added.

“Though it has been suggested that for some ecosystems regime shifts may be slow⁵ regime shifts in lakes and shallow lakes have been characterized as catastrophic and occurring over short time scales⁶.”

The question of the speed of change and drivers of it and reference to Hughes et al 2013 have been included.

I 120-121: why is it notable? and what does it mean?

Rephrased

I 187: what does moot mean?

Moot definition: Subject to debate, dispute, or uncertainty.

I 191: persistent drying maintains persistent or transient dynamics? Clarify

Text added to explain relevance.

I 209-210: related to my point 1 above: gradual change over time does not exclude the existence of thresholds

Caveat added but see earlier point – gradual change/lagged change/ long term equilibrium

I 222-223: does the data presented here confirm the greater plant abundance variation?

No we do not present plant data here – see general comments above, hence the reference to other work

Reviewer #2 (Remarks to the Author):

Davidson et al. use two datasets to test empirically for the existence of alternative stable states in shallow lakes, one a contemporary dataset comprising 902 lakes where they explored relationships between nutrient and chlorophyll a concentrations in the water column, and the other a palaeolimnological analysis of biological communities in three lakes. There is much to like in this paper. The idea being tested is important with impactful implications. There are, however, some issues with the study that mean that the findings are just not sufficiently compelling to convince the reader of their robustness or generality. I think that the authors are on the right track with their efforts, and I hope that they find my comments and recommendations below useful in strengthening this interesting and exciting work.

The contemporary analyses:

The authors utilise a large dataset comprising 902 lakes from North America and Denmark to test for

multimodality in water column chlorophyll a concentrations at this large spatial scale and to explore the strength and nature of relationships between water column nutrient (total P and total N) concentrations and chlorophyll a concentrations. This all seems perfectly reasonable on the face of it, but I think there are a number of things that need additional thought:

- The HD test was used to test for multimodality in the ecological response variable, but why not also use it to test for multimodality in the pressure-ecological response relationship (that is, on the residuals of the nutrient-chl linear model), given that such a bimodal response is predicted for alternative stable states as per the conceptual model depicted on Fig 1?

- The HD test for the 5 year mean was very close to significance (looks like $0.05 < p < 0.1$ on Fig 3), but this actual p-value was not reported in the text and the important implications of this were not discussed. Surely this is a critically important finding in the context of the manuscript? Using mean data averaged over the longer term clearly increases strongly the likelihood of finding bimodality in the data, yet this important finding is just brushed over uncritically. If the authors want to address the stated research questions in a robust and convincing manner, this important finding needs to be addressed head-on in the manuscript and not brushed under the carpet as it does not fit with the narrative of the manuscript, as seems to the reader to have been done here.

See introductory section, HD test removed from the paper and replaced with the Kernel density plot for reasons given in introductory test.

- I have a fundamental issue with the a priori assumption of the conceptual model and simulated data that alternative stable states are to be expected over the entire pressure gradient (in this case, water column nutrient concentrations). Why would we expect similarly strong bimodality in the chl a response at very low and very high nutrient concentrations, as depicted in Figs 1 and 2? I argue that this is in fact not the expectation, and rather that we should instead expect bifurcation in the response above a certain “threshold” on the pressure gradient. In fact, this is exactly what seems to be occurring on the scatterplots of the real-world empirical data shown on Fig 2. Testing explicitly for changes (increases) in the variance of the data over the pressure gradient (analogous to that done by Hillebrand et al., reference [14] in the ms) would address this. My strong recommendation here is that the authors revisit their a priori expectations, simulations and analyses to develop a much more robust and convincing set of tests of their research questions.

We agree with the reviewer that the robust relationship between nutrient and chlorophyll-a along the full length of the enrichment gradient it is a surprising result. We too expected an increase in variance and a concomitant decrease in R^2 as the length of the eutrophication gradient increased, as the chlorophyll-a data became more varied and more unpredictable for a given level of enrichment. However, in particular with the use of the longer-term perspective we did not find this pattern.

In addition, as described in the introductory text above we repeated the analysis on a subset of the data with a reduced range of TP and TN and the diagnostic tests for the presence or absence of alternative equilibria show the same patterns as for the full range of data. The R^2 of the models for the full data range for each temporal grain (1 year, 3 year and 5 year mean) were always higher than for the reduced set, which would not be the case if the variance increased over the pressure gradient. The plots in figure

2 show similar spread in chlorophyll-a at higher concentrations to the simulated data (which we know have no increase in variance over the pressure gradient as we set the parameters).

There are some outliers, e.g. low chlorophyll a for a given TP in the 3 year mean data and also in the 5 year data for TP. This is in fact not a bifurcation but an artefact of using a single nutrient in the plot when both combine to make the pressure gradient. An analysis of the variation in the molar N:P ratio illustrates the cause of the 'outlier' sites (See S1. Supplementary Materials) as in some case chlorophyll-a is much lower than expected for a given TP as TN is low is limiting Chlorophyll-a in this instance. This is why we chose to include both TN and TP in models as both nutrients have different levels of influence along their combined pressure gradient.

- Please note that the analyses need to be reported in the results in the same order as the research questions were posed. This is not the case at present and is confusing for the reader.

Noted and addressed

- Finally, the single year vs multiple year analysis of this dataset is not actually relevant in terms of the hypotheses. I recommend relegating that research question and set of analyses to the supporting information. It adds unnecessary and tangential detail to the manuscript that undermines the central message and reduces the impact of the work.

We suspect there may be some misunderstanding here as from our perspective the comparison single year versus multiple year data is the kernel the study. It is only from a multiple year perspective that that the highly predictable and linear nature of the relationship between the combination of TP and TN and chlorophyll-a becomes evident. The simulations of data with and without alternative equilibria and then applying the same randomized sampling and bootstrapping at single and multiple year perspectives to compare with the real world data is at the core of the contemporary part of the study.

The palaeolimnological analyses:

While I very much appreciate the inclusion of the long-term perspective the inclusion of these data provide, the scale of the analysis in terms of the ability to provide a general overview of ecological responses to eutrophication is lacking, and is very inconsistent with the very broad and general overview provided by the spatial analysis of the contemporary data. Showing detailed responses of just three lakes is just not sufficient to indicate a general lake response. Unfortunately, inclusion of data from many more lakes is really needed to achieve this.

See main point above.

Reviewer #3 (Remarks to the Author):

Nat Comm

My focus is on the paleolimnological data because the modelling is not within my scope of research.

Lake sediments are a great way to assess catastrophic regime shifts because the indicator taxa (in this case Cladocera) should also change abruptly when the habitat shifts between a clear and macrophyte dominated to turbid and phytoplankton dominated system. In this study, lakes that have been historically documented to shift from macrophyte to phytoplankton dominated systems make ideal test sites. I will say that Felbrigg Lake seems an odd choice to include as it has never completely transitioned between so-called alternative states (it appears to currently be in the process of losing its macrophyte cover). The other two study sites have a dashed red line denoting “plant loss”, whereas in Felbrigg a similar line denotes “loss of zooplanktivore”. This may lead to confusion among readers. Overall, the paleolimnological data show that the Cladoceran indicators show gradual, rather than rapid, changes preceding and following a change from clear to turbid conditions. These data help strengthen conclusions reached from the modelling.

We thank the reviewer for this comment, and we agree the palaeolimnological aspects of the work, though perhaps a little at odds with the rest of the study provide a unique insight into how lakes change in response to eutrophication on a decadal scale. In particular, the high resolution cladoceran analysis provides an excellent archive of ecological change, both bottom up and top down driven.

Minor comments:

Line 41. Add comma after “Since its inception”

Done

Line 103. “2986 lakes years” Seems like odd terminology?

Altered

Line 142. “Prior to the modern era and the major development of eutrophication in the three lakes the plant community was dominated by less nutrient tolerant submerged plants including charophytes (all lakes)”

Not really accurate for Lake Sobygaard - only one peak in charophyte oospores at ~83 cm

Altered

Line 232. It is noted that fish kills have been recorded in the sediment record at Felbrigg; however, it is not clear what proxy(ies) indicates the loss of fish.

The loss of perch was described at the time in records held on the estate. In the palaeo record it is indicated by an increase in *Daphnia ephippia* and also a slight recovering of benthic and plant associated taxa. Added to the supplementary text

Line 233. “The change in macrophyte composition in the sediment cores in all three lakes mirrors those observed as a result of nutrient enrichment over time...”

Ok, but this is hard for readers to assess as little info is given on when nutrient enrichment has occurred over time. No mention is given on TP values over time in the 3 lakes. Likewise, on the sediment profile diagrams there are no proxies of lake production (e.g., fossil pigments, TOC).

Reply: A description of the pollution history of Lake Sobygaard is added to the supplementary information and Loss on ignition data, reflecting organic content of the sediment, which provides information on the eutrophication history of the lakes were added to the main figure and described in the supplementary results.

Line 237. "In addition, in the palaeolimnological records analysed here, pelagic taxa dominated the assemblages present prior to the complete total loss of submerged vegetation."
This is true, but in Felbrigg Chydorus dominates much of record but declines towards surface. In Sobygaard, benthic taxa only ever made up a very small portion

Perhaps a good point to stress here is that in all cases with plant loss there was no rapid regime shift in Cladocerans assemblages. In Sobygaard, planktonic Bosmina remained dominant, and in the other two study sites there was a decline in benthic Chydorus and an increase in planktonic Bosmina, but nothing that would constitute a regime shift. Thus, what is observed is gradual, rather than step-changes.

Changed text in the results

Table 1. Better to report TP in more familiar units of $\mu\text{g/L}$, rather than mg/L ?

Changed

- 1 Capon, S. J. *et al.* Regime shifts, thresholds and multiple stable states in freshwater ecosystems; a critical appraisal of the evidence. *Sci Total Environ* **534**, 122-130 (2015).
- 2 Hillebrand, H. *et al.* Thresholds for ecological responses to global change do not emerge from empirical data. *Nat Ecol Evo* **4**, 1502-1509 (2020).
- 3 Spears, B. M. *et al.* Ecological resilience in lakes and the conjunction fallacy. *Nat Ecol Evo* **1**, 1616-1624 (2017).
- 4 Søndergaard, M., Davidson, T. A., Lauridsen, T. L., Johansson, L. S. & Jeppesen, E. Submerged macrophytes in Danish lakes: impact of morphological and chemical factors on abundance and species richness. *Hydrobiologia* (2021).
- 5 Hughes, T. P., Linares, C., Dakos, V., van de Leemput, I. A. & Van Nes, E. H. Living dangerously on borrowed time during slow, unrecognized regime shifts. *Trends in Ecology and Evolution* **28**, 149-155 (2013).
- 6 Scheffer, M. & Carpenter, S. R. Catastrophic regime shifts in ecosystems: linking theory to observation. *Trends Ecol Evol* **18**, 648-656 (2003).

Reviewer comments, second round review –

[Editor's note: Reviewer #1 was not available this time, but we replaced them with a new reviewer of similar expertise]

Reviewer #2 (Remarks to the Author):

The authors have in general dealt well with the reviewer recommendations regarding their revised approach to the analyses of the “contemporary data”, and I am happy to see that the results described in Figure 3 are now really quite convincing.

However, the palaeolimnological analyses as currently included in the manuscript are neither convincing nor robust in terms of supporting the stated aims of the study. Simply, the analysis of paleolimnological data from three lakes is palpably insufficient to claim any sort of general response of shallow lakes, and does not contribute in any meaningful way to our understanding of the general existence (or not) of rapid shifts in the structure/function of lake ecosystems under nutrient enrichment. All that these results show is that some lakes respond relatively gradually to gradually increasing nutrient enrichment. “No evidence of catastrophic shifts” [line 216] in data from three lakes is really quite meaningless in the overall context of what the manuscript is trying to show and really undermines its central message. Using the data from these three lakes appears subjective in terms of the choice of lakes, and leaves the authors open to the accusation of simply cherry-picking these results from these particular lakes. To be convincing, I reiterate my view that data from many many more lakes need to be included in such an analysis. At present, the data contribute absolutely nothing to our understanding of the generality of these responses across shallow lakes and detract considerably from what are now quite convincing analyses of the contemporary data. I recommend strongly that the palaeolimnological component of this manuscript is either removed entirely or else expanded considerably with data from many more lakes in order to be made more robust and provide convincing objective evidence from multi-decadal timescales.

In general, the revision has decreased significantly the flow of the manuscript and the writing and structuring of the work is now really quite poor in places. For example, Figure 1 is not referenced in the main text until the end of the second paragraph of the Results section, after Figure 2 is first cited (it is cited in the Abstract, which is not an appropriate location for a citation to a display item). There is no description of what “contemporary data” actually means and the reader is left to work that out for themselves when the term first appears out of the blue in the Results section. Similarly, the first sentence in the decadal scale subsection in the results–“The multi-indicator palaeolimnological analysis...” also comes out of the blue with absolutely no context or introduction to this work in the Introduction (same for “simulated data sets” [line 132], “the three lakes” [line 176]). The revised legend of Figure 1 is written very poorly and contains some orphan text.

Reviewer #3 (Remarks to the Author):

I had far fewer comments than the other reviewers as mine were restricted largely to the paleolimnological data. All of these comments were adequately addressed and I have nothing further to add.

Reviewer #4 (Remarks to the Author):

This is an interesting discussion of an important topic. The exchange between reviewer 1 and the authors touches on the heart of the contention, and really would make instructive reading for anyone interested in this topic. I did not review earlier rounds of this manuscript, but as requested

by the editor I will focus my comments around these contentions.

The discussion of proving or disproving the theory of alternative stable states in lake ecosystems is challenging. To those trained in formal mathematics, the notion of proof has nothing to do with empirical observation, but sets a rather higher bar. For instance, I believe one could prove that given a system governed by alternative stable state dynamics (assume a particular model, say May's 1977 model), one could always introduce a sufficiently large environmental noise term σ such that the probability of detecting an alternative stable state from any large number N data points under a given protocol would be less than some arbitrarily small value ϵ . (Many variations on that theme are possible as well, such as other unobserved/latent variable(s) that vary across lakes).

Fortunately the main text does not use the terms prove or disprove, and the authors have reasonably demonstrated the central claim:

> ideas of bimodality and alternative equilibria do not help in explaining the patterns in the data

within the context of the lakes they have examined here. Whether or not that justifies a verdict that 'the theory does not stand the test of time' is perhaps another matter, but to me this is more literary device and the authors may be somewhat forgiven if doing seems to overstate the evidence, as much the same charge might be leveled at the rhetoric on the other side of the debate.

I think this work helps demonstrate how much is still not well understood regarding the dynamics and empirical evidence for tipping points in shallow lake ecosystems. I do not expect that it will close the issue, but will surely become a well-cited and productive catalyst to future research and I look forward to seeing it published.

Response to Reviewers

We are again grateful to the reviewers for taking time to read and assess our resubmission. From the start of the process, the comments have been insightful and helped improve the quality of the manuscript. It has been a positive reviewing process.

Reviewer #2 (Remarks to the Author):

The authors have in general dealt well with the reviewer recommendations regarding their revised approach to the analyses of the “contemporary data”, and I am happy to see that the results described in Figure 3 are now really quite convincing.

However, the palaeolimnological analyses as currently included in the manuscript are neither convincing nor robust in terms of supporting the stated aims of the study. Simply, the analysis of paleolimnological data from three lakes is palpably insufficient to claim any sort of general response of shallow lakes, and does not contribute in any meaningful way to our understanding of the general existence (or not) of rapid shifts in the structure/function of lake ecosystems under nutrient enrichment. All that these results show is that some lakes respond relatively gradually to gradually increasing nutrient enrichment. “No evidence of catastrophic shifts” [line 216] in data from three lakes is really quite meaningless in the overall context of what the manuscript is trying to show and really undermines its central message. Using the data from these three lakes appears subjective in terms of the choice of lakes, and leaves the authors open to the accusation of simply cherry-picking these results from these particular lakes. To be convincing, I reiterate my view that data from many many more lakes need to be included in such an analysis. At present, the data contribute absolutely nothing to our understanding of the generality of these responses across shallow lakes and detract considerably from what are now quite convincing analyses of the contemporary data. I recommend strongly that the palaeolimnological component of this manuscript is either removed entirely or else expanded considerably with data from many more lakes in order to be made more robust and provide convincing objective evidence from multi-decadal timescales.

**** We have reluctantly removed the palaeolimnological part of the manuscript. The reluctance comes in parts from the fact that it was the inspiration for the ideas about testing alternative equilibria. However, we can see the resolution and extent of contemporary data and the palaeolimnological data area a bit at odds.**

In general, the revision has decreased significantly the flow of the manuscript and the writing and structuring of the work is now really quite poor in places. For example, Figure 1 is not referenced in the main text until the end of the second paragraph of the Results section, after Figure 2 is first cited (it is cited in the Abstract, which is not an appropriate location for a citation to a display item). There is no description of what “contemporary data” actually means and the reader is left to work that out for themselves when the term first appears out of the blue in the Results section. Similarly, the first sentence in the decadal scale subsection in the results—“The multi-indicator palaeolimnological analysis...” also comes out of the blue with absolutely no context or introduction to this work in the

Introduction (same for “simulated data sets” [line 132], “the three lakes” [line 176]). The revised legend of Figure 1 is written very poorly and contains some orphan text.

**** We have reordered some sections of the results to improve the flow of the data. Figure 1 is now the first figure addressed and the text has a more logical flow.**

Reviewer #3 (Remarks to the Author):

I had far fewer comments than the other reviewers as mine were restricted largely to the paleolimnological data. All of these comments were adequately addressed and I have nothing further to add.

**** We removed the palaeolimnological part, please see our comments to reviewer 2.**

Reviewer #4 (Remarks to the Author):

This is an interesting discussion of an important topic. The exchange between reviewer 1 and the authors touches on the heart of the contention, and really would make instructive reading for anyone interested in this topic. I did not review earlier rounds of this manuscript, but as requested by the editor I will focus my comments around these contentions.

The discussion of proving or disproving the theory of alternative stable states in lake ecosystems is challenging. To those trained in formal mathematics, the notion of proof has nothing to do with empirical observation, but sets a rather higher bar. For instance, I believe one could prove that given a system governed by alternative stable state dynamics (assume a particular model, say May's 1977 model),

one could always introduce a sufficiently large environmental noise term σ such that the probability of detecting an alternative stable state from any large number N data points under a given protocol would be less than some arbitrarily small value ϵ . (Many variations on that theme are possible as well, such as other unobserved/latent variable(s) that vary across lakes).

Fortunately the main text does not use the terms prove or disprove, and the authors have reasonably demonstrated the central claim:

> ideas of bimodality and alternative equilibria do not help in explaining the patterns in the data

within the context of the lakes they have examined here. Whether or not that justifies a verdict that 'the theory does not stand the test of time' is perhaps another matter, but to me this is more literary device and the authors may be somewhat forgiven if doing seems to overstate the evidence, as much the same charge might be leveled at the rhetoric on the other side of the debate.

I think this work helps demonstrate how much is still not well understood regarding the dynamics

and empirical evidence for tipping points in shallow lake ecosystems. I do not expect that it will close the issue, but will surely become a well-cited and productive catalyst to future research and I look forward to seeing it published.

**** No substantive changes requested here. We are grateful to the reviewer for their balanced and positive assessment.**

Reviewer #1:

First let me apologise for the very delayed response. Perhaps my comments are not anymore necessary and I completely understand.

I also would like to apologise to the authors. They have come at great lengths to accommodate my comments and it would be the least impolite not to acknowledge that and respond to them from my side.

So please use the following as you deem most appropriate.

I think the manuscript has become more clear and I am glad some of my comments have been useful. Others have triggered a strong discussion.

I have few remaining concerns to share.

My main concern revolves around the robustness of the presented analysis especially as the authors suggest to be used for spatial datasets to explore alternative states.

First, I still find it hard to follow and there are certain things to be clarified (see part at the end of the document), and second there are some more conceptual concerns, which I outline below.

a. The means estimated in the real lakes come from temporally correlated records (there is some legacy from year to year). In the suggested simulated data this is not the case as all values in a lake are independent temporally (unless I am missing something).

****The means in the real world data are individual values per lake and so whilst it is true that individual summer mean values within the time series may be temporally correlated the means used across years are not. Here, our bootstrap approach prevents temporal correlations from having any effect on the data analysis, making real data and simulated data completely comparable from the temporal dependence viewpoint: In our approach, the most likely case is each lake appearing only once, since at the lake level we sample each lake only once on average (Fig. 4, step 3a.). In the bootstrap recombination, the same lake or even the same data point may appear twice or more times (however, repeated appearance is more unlikely than the average appearance, step 3a and 3b). This is expected and actually desired behaviour of the bootstrap (see e.g. Efron & Tibshirani 1993). However, we recombine the data 1000 times, destroying on purpose any temporal coherence of the data by reshuffling it. Therefore, (a.) the average sample has no temporal coherence, as each lake only appears once, and (b.) reshuffling 1000 times results in the fact that even if single iterations contain some temporal dependence by random, this will not affect the distribution of values around the mean**

Efron, B., Tibshirani, R., 1993. *An Introduction to the Bootstrap*. Chapman & Hall/CRC.

In response to my comment on using a eutrophication model with and without ASS rather the phenomenological approach followed the authors assert that their conclusions would be the same.

****There are simulated data with and without ASS. Therefore, our simulations contain a eutrophication model for both cases, and completely fulfill the reviewer's wish.**

b. If there are shifts between alternative states or the difference between the alternative states is small, then taking means will smooth the data (as it can be seen in fig S1) and a linear model will perform better even if alternative states are present.

****If the difference between alt states is small, are they different? The different constellations of simulated data show how the models perform in the presence, absence and prevalence of alternative states. Moreover, any published approach would have issues finding alternative states or tipping points in the data if they are too small to be found. They are too small to be found, if the state effect or tipping point behavior is smaller than the random variability of the data. This is the main conclusion not of our paper but of Hillebrand et al 2020 in Nature Ecology and Evolution (from the abstract: "We find that threshold transgressions were rarely detectable, either within or across meta-analyses. Instead, ecological responses were characterized mostly by progressively increasing magnitude and variance when pressure increased. Sensitivity analyses with modelled data revealed that minor variances in the response are sufficient to preclude the detection of thresholds from data, even if they are present."). Since there are no ecological data without random noise / without variance, no alternative states or small alternative states with random noise / variance are statistically the same thing.**

Hillebrand, H., Donohue, I., Harpole, W.S., Hodapp, D., Kucera, M., Lewandowska, A.M., Merder, J., Montoya, J.M., Freund, J.A., 2020. Thresholds for ecological responses to global change do not emerge from empirical data. *Nature Ecology & Evolution*. <https://doi.org/10.1038/s41559-020-1256-9>

Specifically in I 688 it is stated: Intercepts and slopes of the simulation, resembled the range of the true data (see scatter plots in Fig. 2 of the main manuscript).

It is unclear how this is estimated and which are two probable ASS in the real data. I am not sure how one could derive that from the data.

Also why change both slope and intercept?

Such choices are important because it is unclear how the differences in intercept and slope will affect the detectability of ASS? If the difference is small would it be able to discriminate still between the two?

****The chosen ASS is the most probable, as we described in detail in the Simulation section of the methods. Please refer to this section, where we argue for each decision we took for the simulated dataset in the main manuscript. However, concerning different slope and intercept, the reviewer is right that we did not argue about why we took this decision and we have now added this explanation to the methods section. We based the selection of the slope and the intercept on Scheffer and Carpenter (2003, please see Figure 3 therein for the diagnostics), which propose "(a) shift in a time series, (b) multimodal distribution of states, and (c) dual relationship to a control factor" (Fig. 3 caption within Scheffer and Carpenter 2003) as diagnostics. Here, the idea is that an ecosystem will jump from one state to the next at the same (nutrient) conditions (different**

intercept, condition a within Scheffer and Carpenter), where any change in the nutrient will have different effect on algae or macrophytes (different slope, condition c within Scheffer and Carpenter), resulting in a multimodal distribution of the response (condition b in Scheffer and Carpenter). Hence, our main manuscript simulation is in line with what is predicted for ASS, but we took great lengths to also show other possibilities with the simulations in the SI to really make sure that we did not miss any occasional occurrence.

Scheffer, M., Carpenter, S.R., 2003. Catastrophic regime shifts in ecosystems: linking theory to observation. *Trends in Ecology & Evolution* 18, 648–656.
<https://doi.org/10.1016/j.tree.2003.09.002>

c. I appreciate that the authors use a set of different scenarios but I think the suggested scenarios do not completely address my original point 5, where a mix of lakes with alternative state and without could be also a plausible scenario. In that case, the relative proportion of the two could determine the response of the diagnostic tests. For instance if 10% of the lakes do have a bistable pattern and the rest not, it might be that the overall signal is linear. In that case relative deviations from strong linearity could infer the probability of alternative states (or outliers) in a mixed dataset.

**** The mixed scenario the reviewer refers to is the ASS simulation scenario of the main manuscript. Here 66% of the lakes have no bi-stability, either following one or the other ASS. Only 33% have both scenarios and, hence, bi-stability within the same time series. Moreover, in scenario 4 in the SI, no time series has bi-stability, following one or the other stable state, and still we identify ASS in the diagnostic tests. We furthermore checked, whether the longer time series in the data show signs of bi-stability (SI). Here, only one case showed signs of bi-stability, which we attribute to TN:TP ratios rather than feedback loops (see also SI). Finally, we cannot exclude the case where only a few percent of the lakes are bi-stable, however, this would mean that ASS are the rare to the point of being undetectable, rather than the rule and actually would support our conclusions that we do not need alternative equilibria to explain long-term patterns of Chla.**

d. Lastly, in the simulated datasets, were there gaps introduced to meet the peculiarities of the real data? Could that affect the results?

**** We cannot think of a case where this would affect the results.**

In addition to the above methodological concerns and clarifications, I think the paper still misses a clear big figure with the raw data from all lakes as function of TP and TN.

Now (if I am not mistaken, see below my point xx), this is a small panel within Fig2 whereas fig S2 shows results for a small selection of lakes.

In fact I am still not sure why it is not enough to focus only on the single years? what would be the expectation? One could use the same rationale of diagnostics (residuals and densities) directly on the total single year data.

The authors state that : "If we took only single year data the results are less clear, which in part explains the plausibility of the idea of alternative equilibria, but the multiple year data provide a clear test which has no a priori likelihood of proving or disproving the existence of alternative states." Could the authors expand why the single year are not good proxy (their simulated data don't disprove that) but that the multiyear data are a clear test? What does clear test mean?

A large part of the discussion is concerned with why the single year perspective is not fit for purpose to test for the existence or prevalence of alternative stable states. There are a range of internal and external factors that cause variation in the nutrient to chlorophyll-a relationship within a single year but which even out over a longer time scale. The cloud of single year data just displays a weak relationship between nutrients and chlorophyll-a which is provide no evidence of the presence or absence of alternative equilibria.

Similarly, in a response to one of my comments that nonlinear models could be tested for the relationship between TP and chl_a for the single years, the authors state that even is nonlinear fit could be better this would not affect the results of the study.

I am sceptical why this would have no effect for the presented results.

First I d claim that it will indicate that the linear model the authors support is not perhaps the best.

Second, it should suggest that another than a linear model should be used for producing the simulated datasets.

****The aim of the paper was to assess how linear models performed with the simulated and real world data. A non-linear model may have performed better for single year data but the point is that the real world data showed the same characteristics as the data simulations where Alternative states were absent. In addition the very large amount of variance explained by the linear models, combined with the absence of patterns in the residuals mean that non-linear models could not have performed better when the longer term data were analysed.**

On the response to my point over the BP in paleo data- this point is based on community composition whereas the lake datasets analysis is based on chl A. How do these two connect along the presented framework of the long timescales to be considered? As I have suggested before, I still find the paleo part could be omitted.

****Now omitted**

Also Fig 1 presents these two approaches as detecting alternative states in spatial data at different temporal scales. You may want to slightly rephrase because I dont think the paleo data analysis refers to the spatial data.

****Now omitted**

Technical clarifications:

If I understand the procedure well, in the 1000 bootstraps choosing the same 99 lakes for the single year comparison, how many unique lakes of the total lakes were actually analysed?

****There were 1000 draws from the data set of 99 lakes with long time series so all 99 lakes were likely drawn from multiple times**

I 628-632: the single years vs multi-year mean for the example of 5 years, means that the single years are those on which the 5 year mean was estimated? This is the meaning of fig 2.

In that same fig2 why the single years are bootstrapped? Is that just all the years from all lakes? The figure legend leaves the impression that this is not the case.

****The single year data are bootstrapped so we can get uncertainty estimates of the range of relationships between nutrients and chlorophyll-a. In the 5 year comparison we only used the 99**

lakes for which there are 5 year mean data as the pool of data for the single year bootstrapping as then we are using the same lakes for the comparison.

I 672: the resulting y_i y_j were randomised using a Gamma distribution. why not adding an error from a gamma distribution on the equations?

**** This is what we did, we rephrase the methods to be clearer.**

. I 694 „appearance of ASS states in the data could happen at any point in the time series of a single lake, or the entire time series could include only one of two alternative stable states“ I dont understand how the alternative states could happen at any point in time as they are generated by a linear model. There are 2 populations, so each timeseries of i or j should give a single state unless mixed. Do you mean these two populations were simulated for 20 years and then they were sampled? I think I don't understand the timeseries meaning.

**** We used two linear models (Pop i and j), either the response of y to x was based on Pop i or j , where the used linear model was decided randomly for the time series. E.g. The Pop i linear model could be used for the first 40% of a time series, and the Pop j linear model could be used for the second 60% of a time series.**

. I 701: What does this mean? Since the variability and range of xx (nutrient) and yy (Chl- a response) is simulated as close as possible to the real-world data in all scenarios (see also “General approach”), the measures taken here (variable time series & combination of different alternative stable state scenario constellations) produce a simulation as close to the real-life data as possible

**** The mean and variability of the nutrient data are based on the mean range and CV of the real TN data, and this variability is the same for the TP data), and the variability of the Gamma distributions are based on the Chl a variability of the real data. The slope and intercept values were chosen to resemble a range which would be expected for true Chl a responses to nutrients, the length of the time series vary in the same range as the real-data time series, and the mix of ASS onsets within time series are our best guess for what to expect for ASS in real data. Therefore our scenarios are as close to the real data as possible. This is now more detailed in the methods of the main text (section “Simulation scenarios based on characteristics of real world data.”).**

I dont understand the bandwidths in the kernel density nor how they are interpreted to conclude alternative states.

**** The bandwidths were chosen to account for lower and higher frequency variability in the density plots. This was to show that neither type of frequency bands reveals clear signs of ASS (no clear multimodality).**